## OPEN

# *Plasmodium falciparum* is evolving to escape malaria rapid diagnostic tests in Ethiopia

Sindew M. Feleke [1,9 ✉], Emily N. Reichert [2,9], Hussein Mohammed [1], Bokretsion G. Brhane[1],
Kalkidan Mekete[1], Hassen Mamo [3], Beyene Petros[3], Hiwot Solomon[4], Ebba Abate[1], Chris Hennelly[2],
Madeline Denton[2], Corinna Keeler [2], Nicholas J. Hathaway[5], Jonathan J. Juliano[2], Jeffrey A. Bailey[6],
Eric Rogier[7], Jane Cunningham[8,10 ✉], Ozkan Aydemir[6,10] and Jonathan B. Parr [2,10 ✉]

In Africa, most rapid diagnostic tests (RDTs) for falciparum malaria recognize histidine-rich protein 2 antigen. *Plasmodium falciparum* parasites lacking histidine-rich protein 2 (*pfhrp2*) and 3 (*pfhrp3*) genes escape detection by these RDTs, but it is not known whether these deletions confer sufficient selective advantage to drive rapid population expansion. By studying blood samples from a cohort of 12,572 participants enroled in a prospective, cross-sectional survey along Ethiopia's borders with Eritrea, Sudan and South Sudan using RDTs, PCR, an ultrasensitive bead-based immunoassay for antigen detection and next-generation sequencing, we estimate that histidine-rich protein 2-based RDTs would miss 9.7% (95% confidence interval 8.5–11.1) of *P. falciparum* malaria cases owing to *pfhrp2* deletion. We applied a molecular inversion probe-targeted deep sequencing approach to identify distinct subtelomeric deletion patterns and well-established *pfhrp3* deletions and to uncover recent expansion of a singular *pfhrp2* deletion in all regions sampled. We propose a model in which *pfhrp3* deletions have arisen independently multiple times, followed by strong positive selection for *pfhrp2* deletion owing to RDT-based test-and-treatment. Existing diagnostic strategies need to be urgently reconsidered in Ethiopia, and improved surveillance for *pfhrp2* deletion is needed throughout the Horn of Africa.

*P*lasmodium falciparum strains that evade diagnosis by RDTs represent a major threat to malaria control and elimination efforts[1,2]. Malaria RDTs detect antigens produced by *Plasmodium* parasites, including *P. falciparum* histidine-rich protein 2 (HRP2), parasite lactate dehydrogenase (LDH) and aldolase. HRP2 has advantages over other biomarkers because of its abundance in the bloodstream, repetitive binding epitopes and falciparum-specificity[3–5]. Most HRP2-based RDTs also exhibit some cross-reactivity to a closely related protein (HRP3)[6]. Of the 345 million RDTs sold annually, HRP2-based RDTs are the predominant malaria diagnostic test, the majority of which are deployed throughout sub-Saharan Africa[7].

Deletion mutations involving the histidine-rich protein 2 and/or 3 (*pfhrp2/3*) genes allow parasite strains to escape HRP2-based RDT detection[8,9]. First described in clinical samples from Peru in 2010, these subtelomeric deletions on chromosomes 8 (*pfhrp2*) and 13 (*pfhrp3*) are frequently large (≥20 kb), encompass multiple genes and are difficult to study using existing methods[8,10]. Improved PCR and serological approaches can be used to increase confidence in deletion prevalence estimates[11–13], but our understanding of the evolutionary history of *pfhrp2/3*-deleted *P. falciparum* is limited and largely informed by analysis of a small number of microsatellite markers[14–16]. Recent genomic analyses have begun to expand our understanding of *pfhrp2/3*-deleted *P. falciparum*[17–19] but continue to be hindered by the challenges of assembling the highly repetitive and paralogous sequences of *P. falciparum* subtelomeres[20].

New tools are needed to support surveillance of *pfhrp2/3* deletions, determine their true prevalence and understand the forces impacting their evolution and spread.

'Diagnostic-resistant' *pfhrp2/3*-deleted parasites have now been observed in multiple sites across Africa. Reports from 2017–2018 prompted calls for urgent surveillance in affected regions, including countries in the Horn of Africa like Ethiopia[14,21–26]. Ethiopia is Africa's second most-populous country, and around 60% of its population is at risk of malaria exposure[27]. *Plasmodium falciparum* infection accounts for the majority of malaria deaths and approximately 70% of all cases[27,28]. RDTs were first introduced in Ethiopia in 2004, and the country's current test–treat–track strategy requires parasitological confirmation either by quality microscopy or RDT before antimalarial treatment[29]. *Plasmodium falciparum–Plasmodium vivax* (HRP2/*Pv*-specific-LDH) combination RDTs are the sole diagnostic test used in most settings. Over the past decade, Ethiopia has achieved remarkable progress in the fight against malaria through strong preventative and case management interventions, including engagement of health extension workers to provide diagnostic services at a local level[29]. Reports of highly prevalent *pfhrp2/3*-deleted parasites in neighbouring Eritrea suggest that these gains could be threatened[14,22]. Rapid assessment of the epidemiology of *pfhrp2/3* deletions in Ethiopia and surrounding regions is required to determine whether a change in malaria diagnostic testing policy is warranted.

[1]Ethiopian Public Health Institute, Addis Ababa, Ethiopia. [2]Institute for Global Health and Infectious Diseases, Department of Medicine, Division of Infectious Diseases and Department of Geography, University of North Carolina at Chapel Hill, Chapel Hill, NC, USA. [3]Department of Microbial, Cellular and Molecular Biology, College of Natural and Computational Sciences, Addis Ababa University, Addis Ababa, Ethiopia. [4]Ministry of Health, Addis Ababa, Ethiopia. [5]Department of Medicine, University of Massachusetts Medical School, Worcester, MA, USA. [6]Department of Pathology and Laboratory Medicine, Warren Alpert Medical School, Brown University, Providence, RI, USA. [7]Division of Parasitic Diseases and Malaria, Centers for Disease Control and Prevention, Atlanta, GA, USA. [8]Global Malaria Programme, World Health Organization, Geneva, Switzerland. [9]These authors contributed equally: Sindew M. Feleke, Emily N. Reichert. [10]These authors jointly supervised this work: Jane Cunningham, Ozkan Aydemir, Jonathan B. Parr. ✉e-mail: sindewm@gmail.com; cunninghamj@who.int; jonathan_parr@med.unc.edu

**Table 1 | Characteristics of study participants and RDT results**

|  | Amhara | Gambella | Tigray | Overall |
|---|---|---|---|---|
| **Participants**, n | 3,879 | 2,335 | 6,357 | 12,572 |
| **Age**, median years (IQR) | 20 (10–28) | 12 (5–19) | 21 (9–37) | 19 (8–30) |
| **Female**, n (%) | 1,492 (38.5) | 1,055 (45.2) | 3,008 (47.3) | 5,555 (44.2) |
| **Location**, n (%) |  |  |  |  |
| Rural | 2,350 (60.6) | 923 (39.5) | 4,445 (69.9) | 7,718 (61.4) |
| Urban | 1,282 (33.0) | 82 (3.5) | 1,609 (25.3) | 2,973 (23.6) |
| Missing | 247 (6.4) | 1,330 (57.0) | 303 (4.8) | 1,881 (15.0) |
| **Fever**, n (%) | 3,607 (93.0) | 2,255 (96.6) | 5,593 (88.0) | 11,455 (91.1) |
| **CareStart RDT**, n (%) |  |  |  |  |
| HRP2+, Pv-LDH+ | 59 (1.5) | 509 (21.8) | 25 (0.4) | 593 (4.7) |
| HRP2+ only | 1,053 (27.1) | 165 (7.1) | 507 (8.0) | 1,725 (13.7) |
| Pv-LDH+ only | 241 (6.2) | 11 (0.5) | 338 (5.3) | 590 (4.7) |
| Negative | 2,518 (64.9) | 1,650 (70.7) | 5,486 (86.3) | 9,654 (76.8) |
| Invalid | 8 (0.2) | 0 (0.0) | 1 (0.0) | 9 (0.1) |
| **SD Bioline RDT**, n (%) |  |  |  |  |
| HRP2+, Pf-LDH+ | 719 (18.5) | 552 (23.6) | 276 (4.3) | 1,547 (12.3) |
| HRP2+ only | 297 (7.7) | 106 (4.5) | 201 (3.2) | 604 (4.8) |
| Pf-LDH+ only | 239 (6.2) | 11 (0.5) | 168 (2.6) | 418 (3.3) |
| Negative | 2,609 (67.3) | 1,665 (71.3) | 5,705 (89.7) | 9,979 (79.4) |
| Invalid | 15 (0.4) | 0 (0.0) | 2 (0.0) | 17 (0.1) |

Pv-LDH, *P. vivax* parasite LDH; Pf-LDH, *P. falciparum* parasite LDH.

Using the largest prospective study of *pfhrp2/3*-deleted *P. falciparum* performed so far, we apply genomic tools to determine the genetic epidemiology of *pfhrp2/3*-deleted *P. falciparum* in sites spanning Ethiopia's borders with Eritrea, Sudan and South Sudan. In this study, based on the World Health Organization's (WHO) *pfhrp2/3* deletion surveillance protocol released in 2018 to encourage a harmonized and representative approach to *pfhrp2/3* deletion surveillance and accurate reporting[30], we confirm deletions using multiple PCR assays[13], an ultrasensitive bead-based immunoassay for antigen detection[12], whole-genome sequencing (WGS)[31,32] and/or molecular inversion probe (MIP) deep sequencing[33]. Using a new MIP panel designed for high-throughput *pfhrp2/3* genotyping, we map and categorize deletion breakpoints and evaluate their flanking regions for evidence of recent evolutionary pressure favouring *pfhrp2/3*-deleted parasites.

## Results

**Study population and RDT results.** A total of 12,572 study participants (56% male, 44% female) between the ages of 0 and 99 years who presented with one or more of symptoms consistent with malaria at 108 health facilities in the Amhara, Tigray and Gambella regions between November 2017 and June 2018 were enroled (Table 1). Median participant age was 19 years (interquartile range (IQR) 8–30). From the same fingerprick, participants were tested with two RDTs, including the routine HRP2/Pv-specific-LDH RDT combination test (CareStart Pf/Pv RDT, Access Bio, catalogue no. RM VM-02571) and the survey HRP2/Pf-specific-LDH RDT (SD Bioline Malaria Ag P.f. RDT, Alere, catalogue no. 05FK90, lot no. 05FDC024A).

Overall, 2,714 (22%) study participants were *P. falciparum*-positive by at least one RDT (any HRP2- or Pf-LDH-positive band); among these, 361 (13.3%, 95% confidence interval (CI) 12.1–14.7) had a discordant RDT profile suggestive of *pfhrp2/3*-deleted *P. falciparum*

infection, which was defined as HRP2-negative by both RDTs but Pf-LDH-positive. Participants with positive or discordant RDT profiles were treated according to national guidelines. Of the study participants, 1,183 (9.4%) were *P. vivax*-positive by the CareStart RDT. Among the 2,714 samples *P. falciparum*-positive by RDT, the northern region of Tigray had the highest proportion of infections with discordant RDT profiles at 140/689 (20.4%, 95% CI 17.5–23.7), followed by Amhara with 211/1342 (15.8%, 95% CI 13.9–17.8) and Gambella with 10/683 (1.5%, 95% CI 0.7–2.8), as shown in Fig. 1.

***Pfhrp2/3* deletion PCR genotyping.** Eight hundred and twenty samples with complete demographic and clinical data from Amhara (n = 524), Tigray (n = 225) and Gambella (n = 71) underwent molecular analysis. These samples were collected from participants with the discordant RDT profile and a subset of participants with other RDT results (Fig. 2 and Supplementary Note). Further analysis was restricted to the 610 samples with >100 parasites per μl to avoid misclassification of *pfhrp2/3* deletions due to low parasitaemia (Extended Data Fig. 1); 176 samples (28.9%) had the discordant RDT profile.

Infection by *pfhrp2/3*-negative parasites was common among these 610 participants when assessed by PCR, with 355 (58%, 95% CI 54–62) lacking detectable *pfhrp2* and/or *pfhrp3*, and 136 (22%, 95% CI 19–26) lacking both *pfhrp2* and *pfhrp3*. For those lacking only one gene, *pfhrp3*-negative infections (192 (31%, 95% CI 28–35) *pfhrp2+/pfhrp3−*) were more prevalent than *pfhrp2*-negative infections (27 (4.4%; 95% CI 3–6) *pfhrp2−/pfhrp3+*). We observed expected agreement between the results of *pfhrp2/3* PCR assays, RDTs and a bead-based HRP2 immunoassay applied to a randomly selected subset of samples (Table 2, Extended Data Fig. 2 and Supplementary Note). No associations between *pfhrp2/3* PCR result and age, sex or parasitaemia were identified (Supplementary Table 1 and Extended Data Fig. 3).

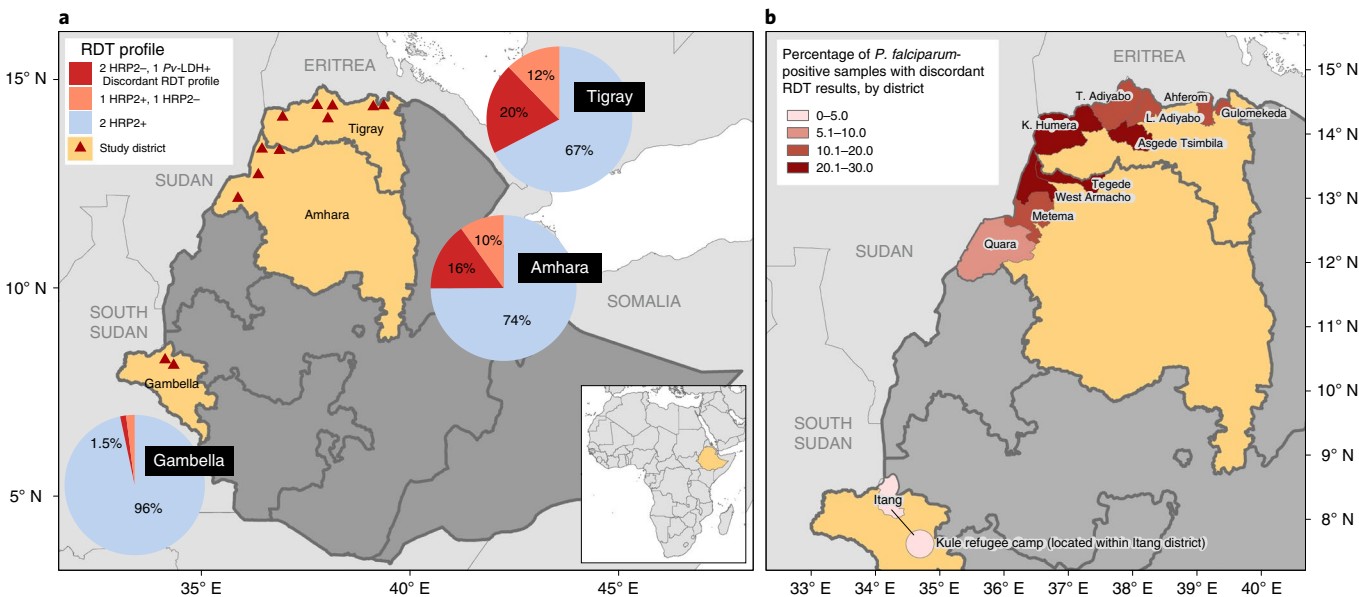

**Fig. 1 | Distribution of *P. falciparum*-positive RDT results and discordant profiles suggestive of *pfhrp2/3* gene deletions. a**, Aggregated results from both RDTs, CareStart *Pf/Pv* (HRP2/*Pv*-LDH) RDT and SD Bioline Malaria Ag P.f. (HRP2/*Pf*-LDH) RDT, displayed by region for all *P. falciparum* infections (*n* = 2,714). The '2 HRP2−, 1 *Pf*-LDH+' discordant RDT profile indicates potential infection by *pfhrp2/3*-deleted *P. falciparum*. Triangles represent the enrolment sites, including 11 districts and the Kule refugee camp within the Itang district in Gambella. **b**, Percentage of study participants identified with *P. falciparum* infection by RDT who had the discordant RDT profile, by district.

***Pfhrp2/3* deletion prevalence estimates.** Incorporating RDT and PCR results, we estimated that 9.7% (95% CI 8.5–11.1) of all *P. falciparum* infections across all study sites would have false-negative HRP2-based RDT results owing to *pfhrp2* deletion. Regional prevalence of false-negative RDTs due to *pfhrp2*-deleted parasites varied, with the highest estimates in Tigray (14.9%, 95% CI 12.5–17.7), followed by Amhara (11.5%, 95% CI 9.8–13.4) and Gambella (1.1%, 95% CI 0.6–2.0). Our prevalence estimates include only samples with both the discordant RDT profile and a *pfhrp2*-negative call by PCR. Parasites with a deletion of *pfhrp2* but intact *pfhrp3* and sufficient cross-reactive HRP3 to trigger a positive HRP2 band on either RDT are not included in these estimates. Thus, the estimated prevalence of false-negative RDT results caused by *pfhrp2* deletions likely underestimates the true prevalence of *pfhrp2*-deleted parasites in this study.

***Pfhrp2/3* deletion characterization using MIP sequencing.** To enable mapping of *pfhrp2/3* deletion regions and population genetic analyses in large-scale epidemiological studies, we developed a targeted panel of 241 MIPs for highly multiplexed deep sequencing of *pfhrp2*, *pfhrp3* and flanking genes on chromosomes 8 and 13. Among *P. falciparum* PCR-positive samples collected from 926 participants and subjected to MIP capture and sequencing, 375 (40.5%) had sufficient depth-of-coverage to make high-confidence calls (Fig. 3). Analysis of variant-called MIP sequences confirmed mixed infections with complexity of infection ≥2 in only 45 (12%) participants; the majority (*n* = 330, 88%) were infected by a single *P. falciparum* strain. We compared RDT, PCR, HRP2 immunoassay, WGS and MIP sequencing results. Although differences between these approaches were apparent and expected due to differences in targets and methodologies, there was strong concordance overall between assays (Table 2, Fig. 4, Extended Data Fig. 4 and Supplementary Note).

***Pfhrp2/3* deletion breakpoint profiling.** Compared with PCR, bead-based immunoassay or RDT diagnosis, MIP sequencing was unique in its ability to reveal distinct subtelomeric structural

profiles along chromosomes 8 and 13 into which samples could be categorized: three for *pfhrp2*+ samples (chr8-P1, chr8-P2 and chr8-P3), one for *pfhrp2*− (chr8-P4), one for *pfhrp3*+ (chr13-P1) and three for *pfhrp3*− (chr13-P2, chr13-P3 and chr13-P4) (Fig. 3). All *pfhrp2*− samples had the same subtelomeric structural profile (chr8-P4), although two other subtelomeric deletions were identified on chromosome 8 that did not involve *pfhrp2* (chr8-P2, chr8-P3). These deletions involve members of the *rifin* and *stevor* gene families, as well as genes of unknown function.

The structural profile of most samples identified as *pfhrp3*− (chr13-P3 and chr13-P4) differed in the presence or absence of a segment of chromosome 13 directly telomeric to *pfhrp3* (position 2,852,540 to 2,853,533) encoding a member of the acyl-coA synthetase family (PF3D7_1372400). All chr13-P3 and chr13-P4 deletions resulted in loss of genes with roles in red blood cell invasion (PF3D7_1371700, serine/threonine kinase and member of the FIKK family; PF3D7_1371600, erythrocyte binding-like protein 1 (*EBL-1*))[34,35], whereas they were present in all chr13-P1 (*pfhrp3*-intact) parasites. The chr13-P2 deletion profile was observed in only one sample from Amhara's Metema district. We did not observe an association between subtelomeric structural profile and the number of symptoms experienced by participants, geographic region or *P. vivax* co-infection (Extended Data Fig. 5, Supplementary Table 4 and Supplementary Note).

Analysis of 25 genomes from *P. falciparum* samples collected in Ethiopia in 2013 and 2015 and available in the MalariaGEN database (Extended Data Figs. 6 and 7) uncovered chromosome 13 subtelomeric structural profiles similar to those identified by MIP sequencing: 9 samples with coverage consistent with chr13-P3 (*pfhrp3*-deleted), 2 samples with chr13-P4 (*pfhrp3*-deleted) and 14 samples with chr13-P1 (*pfhrp3*-intact)[19].

**Genetic signatures of evolutionary selection.** Extended haplotype homozygosity (EHH) statistics revealed signatures of recent positive selection in the flanking region centromeric to *pfhrp2* deletions on chromosome 8 but not in flanking regions around *pfhrp3* deletions on chromosome 13 (ref. [36]). EHH remained very high for parasites

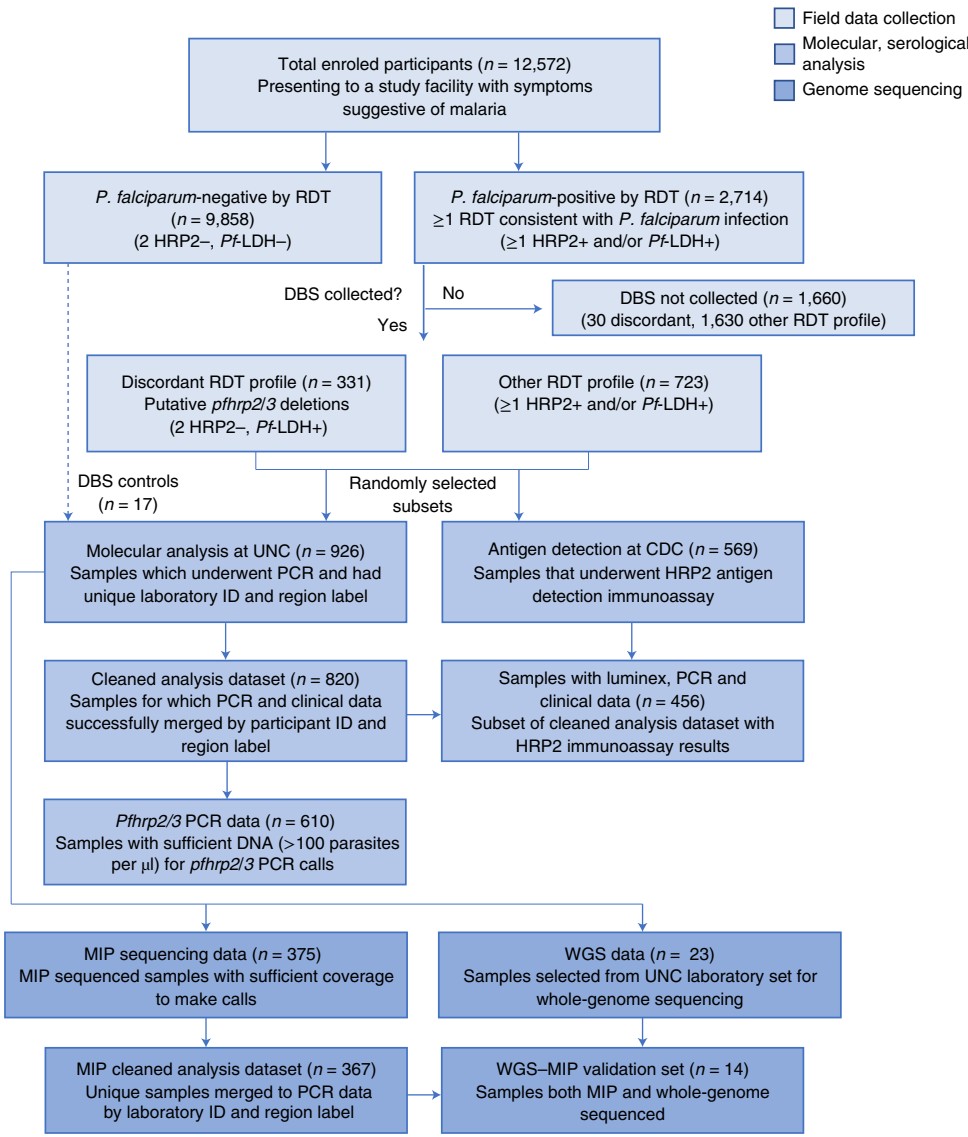

**Fig. 2 | Study samples and assays performed.** Samples were randomly selected by EPHI for molecular and antigen assays and for sequencing to expedite analyses.

with the *pfhrp2* deletion (0.968) along the entire 28 kb analysed, whereas homozygosity around the *pfhrp2*-intact (wild-type) allele quickly broke down (Fig. 5a). A similar pattern was observed when deletion profiles were analysed separately; chr8-P4 EHH remained high and chr8-P1-P3 EHH broke down quickly (Extended Data Fig. 8). We further confirmed high EHH around the *pfhrp2* deletion allele using WGS data. Comparing 23 whole-genome sequenced samples from this study and the 25 published MalariaGEN samples described above, we were able to extend our analysis and confirm an EHH length of >143 kb centromeric to the deletion (Extended Data Fig. 9a). These findings suggest a recent selective sweep, indicative of strong evolutionary pressure favouring *pfhrp2*-deleted *P. falciparum* parasites.

A different pattern was observed in the regions flanking *pfhrp3* (Fig. 5b). EHH quickly decreased below 0.5 for *pfhrp3* deletion alleles as well as the *pfhrp3*-intact allele within 1 kb of available single-nucleotide polymorphisms. When deletion profiles were analysed as separate alleles, the EHH pattern was similarly low for chr13-P1, -P3 and -P4 (Extended Data Fig. 10). Comparison of EHH around the *pfhrp3*-intact and P3-like *pfhrp3* deletions using

WGS data from the 25 MalariaGEN samples confirmed our finding that the EHH decreased quickly for both *pfhrp3*-intact and pfhrp3 deletion alleles (Extended Data Fig. 9b). Taken together, these findings suggest that each *pfhrp3* deletion profile arose multiple times independently, and/or they have been present in the parasite population for sufficient time for homozygosity due to genetic hitchhiking to be degraded by recombination with different haplotypes.

## Discussion

Using the largest prospective study of *pfhrp2/3*-deleted *P. falciparum* performed so far and complementary molecular, immunological and sequencing assays, we provide clear evidence that *pfhrp2/3*-deleted parasites are circulating in multiple sites along Ethiopia's borders with Sudan and Eritrea. Analysis of flanking haplotypes suggests that the *pfhrp2* deletion mutation emerged and recently expanded from a single origin, whereas *pfhrp3* deletion mutations have existed for a longer time span and likely have multiple origins. As expected, we did not observe perfect concordance between RDT results, PCR, a bead-based immunoassay, WGS and MIP sequencing results. However, the preponderance of evidence

## Table 2 | Assay results

| | RDT results | | |
|---|---|---|---|
| | 2 HRP2+, 1 Pf-LDH+ | 2 HRP2−, 1 Pf-LDH+ | 1 HRP2+, 1 HRP2− |
| **PCR**, n (%) | 379 | 176 | 47 |
| pfhrp2+/3+ | 210 (55) | 20 (11) | 19 (41) |
| pfhrp2−/3− | 9 (2) | 114 (65) | 10 (22) |
| pfhrp2−/3+ | 7 (2) | 14 (8) | 5 (11) |
| pfhrp2+/3− | 152 (40) | 28 (16) | 12 (26) |
| **HRP2 immunoassay**, n (%) | 243 | 167 | 42 |
| HRP2+ | 224 (92) | 40 (24) | 30 (71) |
| HRP2− | 19 (8) | 127 (76) | 12 (29) |
| **MIP**, n (%) | 198 | 104 | 30 |
| pfhrp2+/3+ | 77 (39) | 5 (5) | 5 (17) |
| pfhrp2−/3− | 10 (5) | 84 (81) | 10 (33) |
| pfhrp2−/3+ | 3 (2) | 1 (1) | 4 (13) |
| pfhrp2+/3− | 108 (55) | 14 (13) | 11 (37) |
| **WGS**, n (%)[a] | 0 | 22 | 0 |
| pfhrp2+/3+ | NA | 0 (0) | NA |
| pfhrp2−/3− | NA | 22 (100) | NA |
| pfhrp2−/3+ | NA | 0 (0) | NA |
| pfhrp2+/3− | NA | 0 (0) | NA |

PCR, bead-based antigen immunoassay, MIP deep sequencing and WGS for samples *P. falciparum*-positive by RDT are shown. [a]Zero median WGS coverage across 1,000 bp windows encompassing *pfhrp2* or *pfhrp3*, in samples with clinical data. NA, not applicable.

from these diverse platforms provides robust confirmation of deletions and supports use of the WHO protocol for rapid *pfhrp2/3* deletion surveillance[30]. This protocol provides standardized data collection, fieldwork and sampling methods that can be adapted to the local context and is intended to help programmes rapidly determine whether deployment of alternative diagnostics is needed. The prevalence of false-negative HRP2-based RDT results due to *pfhrp2* deletions is estimated at 9.7% overall and up to 11.5% and 14.9% in the Amhara and Tigray regions, respectively.

These estimates exceed WHO minimum criteria (>5%) for a change in national diagnostic testing strategy. *Pfhrp2/3*-deleted parasites threaten recent progress made by Ethiopia's malaria control and elimination programme, and raise concerns about the ongoing use of and exclusive reliance on HRP2-based RDTs in the region for falciparum malaria diagnosis. However, transition to alternative combination diagnostics is not straightforward given the poor performance of *Pf*-LDH-based RDTs during multiple rounds of WHO product testing and versus PCR in this study (Supplementary Note and Supplementary Table 11) and the challenges of conducting high-quality microscopy in the field[37]. Currently, only one combination *Pf*-LDH/*Pv*-LDH product suitable for Ethiopia is approved for purchase using Global Fund financing (https://www.theglobalfund.org/media/5891/psm_qadiagnosticsmalaria_list_en.pdf?u=636438486100000000).

Eritrea's alarming reports of false-negative RDTs due to *pfhrp2/3*-deleted parasites prompted an immediate change in national diagnostic testing policy in 2016[14,38]. Recent evidence from Sudan, Djibouti and Somalia suggests that the Horn of Africa may already be heavily affected by *pfhrp2/3*-deleted parasites[39,40], although results from ongoing surveillance efforts are not yet publicly available. Within affected regions in Ethiopia, we observed spatial heterogeneity in *P. falciparum* RDT profiles by district, with

prevalence of the discordant HRP2−, *Pf*-LDH+ RDT profile ranging from 1 to 30% (Supplementary Table 2). Although finer scale spatial analyses were not possible because of our health facility sampling approach, this finding is consistent with prior studies showing variation within countries and by region[23]. Differences in transmission intensity, treatment-seeking behaviour, diagnostic testing capacity and seasonality may account for some of the spatial variation in *pfhrp2/3* deletion prevalence estimates[16,41–43]. Although the factors driving emergence of these parasites in some regions but not others remain poorly understood, our study suggests that *pfhrp2*-deleted parasites may have spread widely within Ethiopia from a single origin. This finding is consistent with early microsatellite analysis of *pfhrp2/3*-deleted strains in Eritrea, in which 30 of 31 (96.8%) *pfhrp2*-deleted strains fell into a single genetically related cluster[14], and raises concern about clonal expansion of *pfhrp2*-deleted strains in the Horn of Africa.

Using a multifaceted approach, we validate the use of MIP sequencing for high-throughput *pfhrp2/3* deletion genotyping, deletion profiling and population genetic analysis. Comparison of MIP sequencing with other approaches demonstrated that it can be used for cost-effective (approximately US$10–15 per sample) and scalable deletion genotyping in samples with parasite densities of approximately 1,000 parasites per µl. Although this threshold can probably be improved by additional sequencing of samples with inadequate sequencing depth-of-coverage, in this case, the equivalent of half a NextSeq 550 flow cell enabled visualization of deletion breakpoint regions and variant calling in *P. falciparum*'s subtelomeres in a large portion of samples, without the need for costly enrichment and WGS.

Based on analysis of MIP sequencing and available WGS data, we posit one potential model by which *pfhrp2/3*-deleted parasite populations may have evolved in the Horn of Africa. Findings from this study suggest that parasite populations with *pfhrp3* deletions expanded in the more distant past and potentially arose multiple times independently, based on low EHH surrounding *pfhrp3*, multiple deletion profile patterns, the high overall frequency of *pfhrp3*-deleted parasites and their presence in older samples from 2013 in the MalariaGEN study. In this milieu, recent strong selection favouring parasites with deletions of *pfhrp2* probably occurred due to 'test–track–treat' policies that rely on HRP2-based RDTs and allow parasites with deletions of both genes, or in some cases one of the two genes, to escape treatment. Implicit in this model is the assumption that forces apart from RDT-derived pressure are also driving the evolution of *pfhrp2/3* deletions. Malaria due to *pfhrp2+/3−* parasites should be detectable by HRP2-based RDTs[44]. Further, *pfhrp2/3*-deleted parasites are highly prevalent in South America[23], where RDT-based treatment decisions have never been common.

What other advantages might *pfhrp2/3*-deleted parasites have over those with intact genes? Our limited understanding of the biology of these deletions makes this question hard to answer. Several lines of enquiry may be relevant. (1) They may be better adapted to low transmission intensity settings than other strains. *Pfhrp2/3*-deleted parasites appear to be more common in regions with lower transmission and, presumably, lower complexities of infection, as in the current study[42]. However, this trend could also be an artefact of the assays used to detect them, that is, neither PCR, antigen immunoassays nor common sequencing methodologies are well suited to detect a *pfhrp2/3*-deleted strain when *pfhrp2/3*-intact strains have co-infected a human host. (2) Loss of *pfhrp2/3* or flanking genes may alter parasite virulence. Evidence is accumulating that HRP2 plays a role in cerebral malaria and endothelial inflammation during severe malaria[45,46]. People infected by *pfhrp2/3*-deleted parasites may have less severe disease and therefore be less likely to seek treatment, increasing the likelihood of onward transmission. However, we cannot exclude the possibility that *pfhrp2/3* are lost as

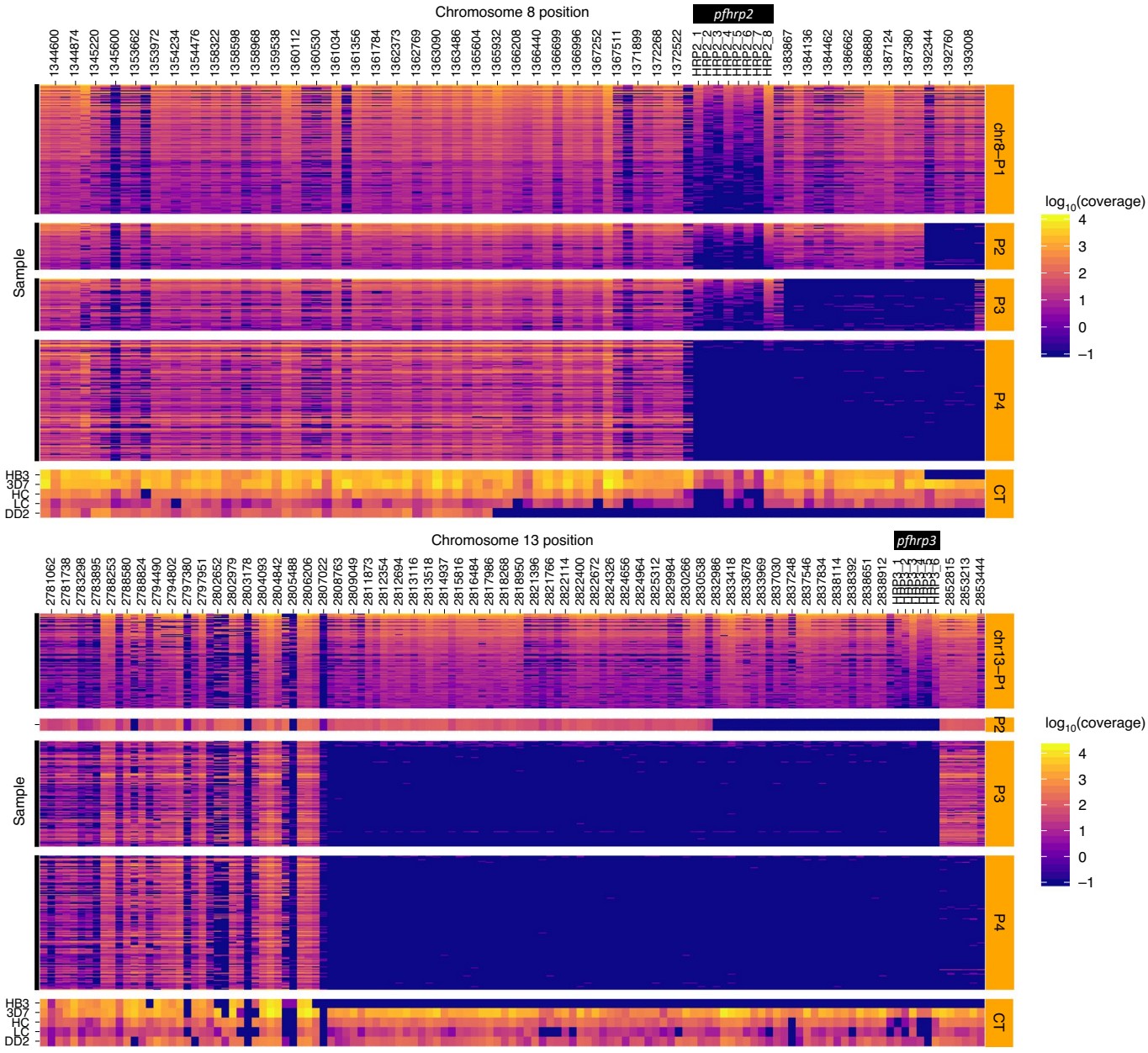

**Fig. 3 | Deletion profiling using MIP sequencing of *pfhrp2* (chromosome 8), *pfhrp3* (chromosome 13) and flanking regions applied to 375 field samples.**
Samples are grouped by subtelomeric structural profile, with control strains denoted CT, as labelled along the right *y* axis. LC (low concentration) and HC (high concentration) controls include mixtures of 1% HB3, 10% DD2, 89% 3D7 strains at densities of 250 and 1,000 parasites per μl, respectively. Columns represent each MIP target segment, rows represent individual samples and the colour scale represents log₁₀(UMI) depth-of-coverage at each location. Columns are labelled by the midpoint of each probe's target region.

a consequence of selection on other genes. For example, the flanking gene *EBL-1* is almost uniformly lost in *pfhrp3*-deleted parasites in this cohort and appears to play a role during invasion of red blood cells[35,47]. Similarly, members of the *rifin* and *stevor* gene families with potential roles in parasite virulence were lost in the subtelomeric deletions observed in this study[48,49]. We did not observe evidence of an association between virulence and subtelomeric deletions in our cohort, but limited clinical data prevents us from assessing the hypothesis rigorously. (3) Loss of *pfhrp2/3* or flanking genes may improve transmissibility to or from mosquitoes. To our knowledge, this phenomenon has not been studied. These and other hypotheses require experimental and improved epidemiological analyses. Regardless of the evolutionary forces at play, our findings strongly

suggest that the evolution of *pfhrp2/3*-deleted parasites in Ethiopia was a multistep process that involved earlier expansion of *pfhrp3*- than *pfhrp2*-deleted parasite populations.

This study has several limitations. First, the study design prioritized evaluation of samples with discordant RDT results (HRP2− but *Pf*-LDH+) for rapid assessment of false-negative RDTs due to *pfhrp2/3* deletions in the context of clinical treatment. This feature of the WHO protocol is intentional because it captures clinically important *pfhrp2/3* deletions and enables real-time, efficient signalling to malaria control programmes of a potential problem. However, it also introduces selection bias that requires careful consideration when estimating the true prevalence of *pfhrp2/3*-deleted parasites. We overcame this limitation by using a conservative

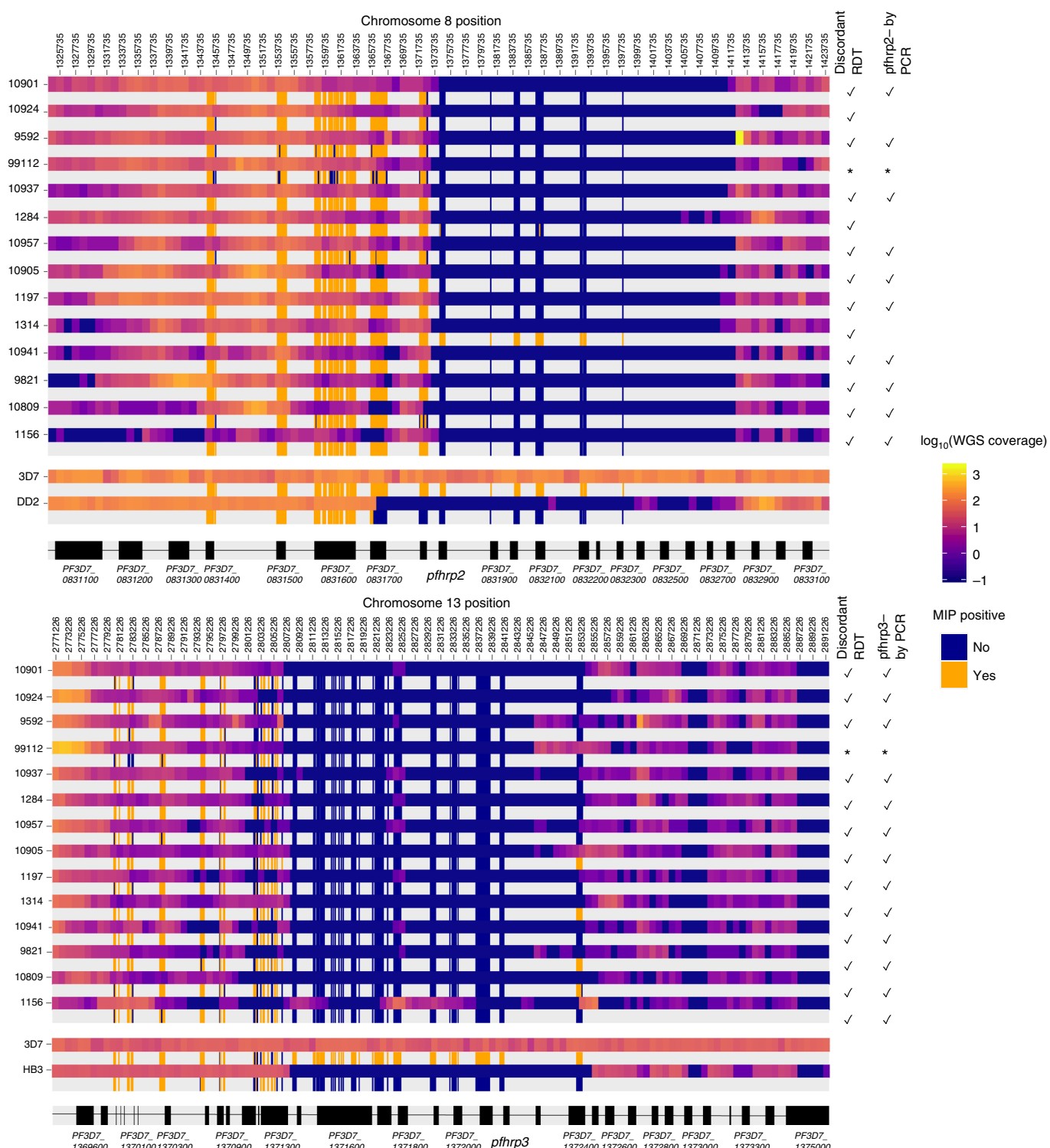

**Fig. 4 | Comparison of MIP and WGS *pfhrp2/3* deletion calls and breakpoint regions.** Among the 14 clinical samples subjected to both methods, each sample is represented by two adjacent rows representing WGS (top) and MIP (bottom) coverage results. WGS coverage is displayed as the log₁₀ median number of aligned reads per 1 kb window. MIP results are coloured by whether each probe captured its target, with intervening regions not targeted in the MIP panel uncoloured. Sample numbers (lab_ID) are provided at left. The locations of *pfhrp2*, *pfhrp3* and flanking genes are shown in black with non-genic regions in grey. The asterisk indicates that it could not be matched with PCR and RDT data.

approach that incorporated both RDT and PCR data to estimate false-negative RDT results due to *pfhrp2* deletions. This metric is relevant to control programmes, but does not capture asymptomatic or low parasite-density infections by *pfhrp2/3*-deleted parasites.

Second, only a subset of samples underwent advanced analysis to expedite reporting, and clinical data were not available for all participants. This was not unexpected for a pragmatic field study of this size. We do not believe that it introduced sufficient bias into

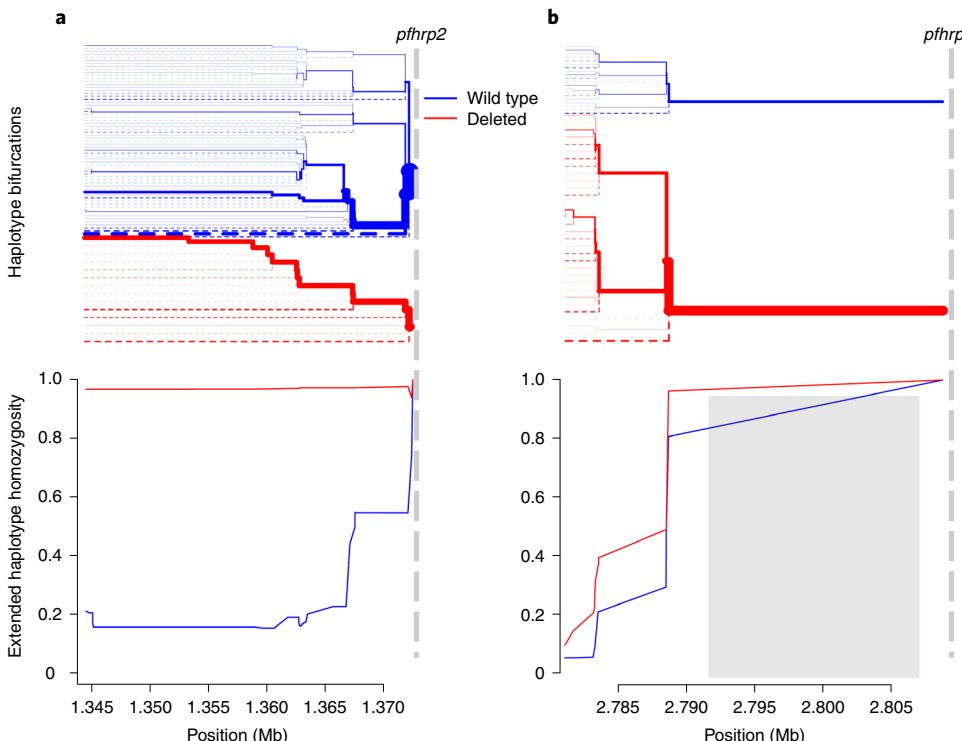

**Fig. 5 | Extended haplotype homozygosity and bifurcation diagrams. a,b** Extended haplotype homozygosity (bottom) and the bifurcation diagrams showing haplotype branching (top) centromeric to the *pfhrp2* (**a**) and *pfhrp3* (**b**) deletions based on MIP data. Vertical dashed lines indicate the centromeric end of deletions. No variant calls were made within the 15.5 kb region on chromosome 13 which is duplicated on chromosome 11, demarcated by the grey box (**b**). Mb, megabase.

our prevalence estimates or population genetic analysis to change our conclusions. Third, we cannot comment on changes in selection pressure over time because the study was cross-sectional. Fourth, we only sampled three regions of Ethiopia, which is a diverse and populous country. The Federal Ministry of Health is now conducting a country-wide survey that will enable comparison of *pfhrp2/3* deletions over time in select sites.

Our large prospective study, using established molecular and antigen detection methods, and a targeted sequencing approach show that *phrp2/3*-deleted *P. falciparum* is a common cause of false-negative RDT results in two regions of Ethiopia. Concerningly, these genomic tools reveal evidence of recent, strong selection for *pfhrp2* deletion in the regions sampled. The selective pressures favouring *pfhrp2*-deleted parasites appear to have occurred on a background of pre-existing *pfhrp3* deletions. Existing malaria control programmes in the region are threatened by expansion of these parasite strains, and prevalence estimates in Tigray and Amhara exceed WHO-recommended thresholds for RDT change. Surveillance has already informed decisions to deploy alternative malaria diagnostics in Eritrea and Djibouti and is underway in Somalia and Sudan. Urgent attention to these deletion mutations is needed to inform malaria diagnostic testing policies throughout the Horn of Africa.

## Methods

**Study design and data collection.** We performed a cross-sectional, multisite study in 11 districts along Ethiopia's borders with Eritrea, Sudan and South Sudan, located within three of its nine administrative regions. On average, ten health facilities were selected from each district, including four districts of Amhara Region (northwest Ethiopia), six districts of Tigray Region (north Ethiopia) and one district of Gambella region (southwest Ethiopia) during the 2017–2018 peak malaria transmission season (September–December, although enrolment in Gambella was completed in April 2018) (Fig. 1). Per WHO protocol[30], each facility passively

enroled participants presenting with symptoms of malaria (fever, headache, joint pain, feeling cold, nausea and/or poor appetite), with sample size proportionally allocated to each facility based on the previous year's malaria case load. All participants provided informed consent, participated in interview questionnaires, underwent blood collection for RDT testing using two types of RDT and were treated according to national guidelines. Data were double-entered into Epi Info (v.3.2), and discrepancies resolved using original paper forms by consensus. Ethical approval was obtained from the Ethiopia Public Health Institute (EPHI) Institutional Review Board (IRB; protocol EPHI-IRB-033-2017) and WHO Research Ethics Review Committee (protocol ERC.0003174 001). Processing of de-identified samples and data at the University of North Carolina at Chapel Hill (UNC) was determined to constitute non-human subjects research by the UNC IRB (study 17-0155). The study was determined to be non-research by the Centers for Disease Control (CDC) and Prevention Human Subjects office (0900f3eb81bb60b9). Experiments were performed in accordance with relevant guidelines and regulations.

**Field sample evaluation.** Study participants were evaluated using both a CareStart *Pf/Pv* (HRP2/*Pv*-pDH) RDT (Access Bio, catalogue no. RM VM-02571) and an SD Bioline Malaria Ag P.f. (HRP2/*Pf*-LDH) RDT (Alere, catalogue no. 05FK90). For the CareStart RDT, 5 µl of capillary whole blood was collected by fingerprick and transferred to the RDT sample well, along with 60 µl of buffer solution. Results were read at 20 min. The SD Bioline RDT followed the same protocol, but with four drops of buffer added and the results read in a 15–30 min window. Participants testing positive by either RDT were first prescribed treatment, according to Ethiopian national guidelines[50].

Cases with any positive HRP2 or *Pf*-LDH RDT band were considered positive for *P. falciparum* malaria. Cases that were *Pf*-LDH-positive but HRP2-negative on both RDTs were considered potential candidates for *pfhrp2/3* gene deletion and defined as 'discordant'. These participants, along with a subset of HRP2-positive and HRP2-negative controls, provided further informed consent for additional blood collection for dried blood spot (DBS) preparation. At least two DBS samples (50 µl per spot) were collected on Whatman 903 protein saver cards (GE Healthcare; catalogue no. 10534612) from consenting participants. DBS samples were stored in plastic bags with desiccant. A randomly selected subset of DBS samples was sent for molecular analysis to UNC and for serological analysis to the CDC.

**DNA extraction and PCR assays.** DNA was extracted from three 6 mm punches per DBS sample using Chelex-100 (Bio-Rad, catalogue no. 1422822)

and saponin (MilliporeSigma, catalogue no. 47036-250G-F) as described previously[51]. Quantitative PCR (qPCR) assays were first performed in duplicate for the *P. falciparum* lactate dehydrogenase gene (*pfldh*)[52]. To avoid the risk of misclassification due to DNA concentrations below the limit of detection for *pfhrp2/3* PCR assays, further analysis was restricted to samples with >100 parasites per µl by qPCR (Extended Data Fig. 1)[13]. PCR assays targeting exon 2 of *pfhrp2* and *pfhrp3* were then performed in duplicate as described previously[6], except that PCR reactions were performed as single-step, 45-cycle assays, using 10 µl of template and AmpliTaq Gold 360 Master Mix (Thermo Fisher Scientific, catalogue no. 4398813) in a 25 µl reaction volume. In addition to no-template and *P. falciparum* 3D7 strain (*pfhrp2+/3+*) positive controls, *pfhrp2* assays included an additional DD2 strain (*pfhrp2−/3+*) control and *pfhrp3* assays included an additional HB3 strain (*pfhrp2+/3−*) control. Finally, an additional single-copy gene, real-time PCR assay targeting *P. falciparum* β-tubulin was performed to confirm that sufficient parasite DNA remained in samples with a negative *pfhrp2/3* PCR result[13]. *Pfhrp2/3* genotyping calls were made in samples with *pfldh* qPCR parasitaemia >100 parasites per µl to avoid misclassification in the setting of amplification failure due to low target DNA concentration. A *pfhrp2* or *pfhrp3* positive call required one or more replicate with distinct band(s) with the expected fragment length. A negative call required both *pfhrp2* or *pfhrp3* replicates to be negative. Detailed reaction conditions for all PCR assays are described in the Supplementary File.

**Serological assays.** The presence of HRP2, pan-LDH and aldolase antigenaemia was assessed in a subset of DBS samples (single 6 mm punch) using a multiplex bead-based immunoassay exactly as described previously[12]. Within this multiplex assay, capture and detection antibodies against the HRP2 antigen would also recognize similar epitopes on the HRP3 antigen, so unique signals for these two antigens cannot be obtained.

**Prevalence estimates.** We estimated the prevalence of *P. falciparum* infections expected to have false-negative HRP2-based RDT results due to *pfhrp2* deletions as follows. First, we calculated the proportion of all RDT-positive *P. falciparum* cases (HRP2+ or *Pf*-LDH+ on any RDT) with the discordant RDT profile (HRP2− on both RDTs, but *Pf*-LDH+), overall and by region. Second, we calculated the observed concordance between the discordant RDT profile and a *pfhrp2*-negative PCR call, overall. Prevalence estimates and 95% CIs were then back-transformed overall and by region using the ci.impt function within the asbio R package (v.1.5-5), which generates CI values for the product of two proportions using delta derivation. This allowed us to estimate with confidence the proportion of *P. falciparum* infections with both *pfhrp2* deletions and false-negative HRP2-based RDT results, overall and by region. As a sensitivity analysis, we also estimated the proportion of those with a discordant RDT and a *pfhrp2*-negative PCR call (directly multiplying the true proportion of *P. falciparum*-positive individuals with a discordant RDT profile, overall and by region, by 0.727, or the overall proportion of discordant RDT samples that had a *pfhrp2−* PCR result). The 95% CI values were then generated using bootstrapping (1,000 iterations). The prevalence estimates and CI values generated by the two approaches were similar (Supplementary Table 3).

**Pfhrp2/3 molecular inversion probe (MIP) development.** *Pfhrp2, pfhrp3* and the flanking regions within a 100 kb window surrounding each gene were targeted for MIP designs using MIPTools[53]. A tiled design strategy was employed that involved multiple, overlapping probes spanning each gene target. Twenty-two genes flanking *pfhrp2* and 31 genes flanking *pfhrp3* were used in the design, of which 11 and 19 were successful on the first design try, respectively. A second attempt was not made for designs for the flanking genes. A total of 241 probes were designed: 9 for *pfhrp2*, 9 for *pfhrp3* and 223 for the flanking genes. MIPs were designed using the 3D7 (v.3) reference genome avoiding hybridization arms in variant regions when possible. Eighty alternative probes accommodating potential variants in the highly variable *pfhrp2* and *pfhrp3* genes were also created. A 15.5 kb segment centromeric to *pfhrp3* on chromosome 13 between positions 2,792,000 and 2,807,500 is duplicated on chromosome 11 between positions 1,918,007 and 1,933,488, with 99.4% sequence identity. Therefore, the target genes falling into this region were multicopy genes and their probes were designed to bind to both loci on the genome (see Supplementary Table 5 for the design overview including all genes targeted, MIPs designed and genomic coordinates). Probes were ordered from Integrated DNA Technologies as 200 pmol ultramer oligos. Probe sequences are provided in the Supplementary Table 6.

**MIP capture and deep sequencing of clinical samples.** All DNA samples extracted by UNC underwent MIP capture using the capture and amplification methods exactly as described by Verity et al.[54], with the exception of oligonucleotides (the *pfhrp2/3* MIP oligonucleotide panel described above was used) and controls (we selected a different set of controls that are informative for *pfhrp2/3* deletion characterization). All MIP captures included multiple controls: 3D7 (*pfhrp2+/3+*), DD2 (*pfhrp2−/3+*), HB3 (*pfhrp2+/3−*) laboratory strains; as well as low- and high concentration mixes (1% HB3, 10% DD2, 89% 3D7) at densities of 250 and 1,000 parasites per µl, respectively. Samples were sequenced on the Illumina NextSeq 550 instrument using 150 bp paired-end sequencing and dual indexing.

**Subtelomeric profiling and variant calling with MIP data.** Read mapping and variant calling were carried out using MIPTools (v.0.19.12.13)[53]. MIPTools uses the MIPWrangler algorithm (v.1.2.0)[55] to create high-quality consensus sequences from sequence read data utilizing unique molecular indexes (UMIs) of MIPs, maps those sequences to the reference genome using bwa (v.0.7.17) and removes off-target sequences as described previously[33,54]. Deletion calls were limited to samples that had high coverage to avoid false positives. Considering the high frequency of large deletions present in the sample set, the coverage threshold was based on a subset of probes that were present on >60% of the samples, none of which overlapped with the chromosome 8 or 13 deletions. Samples with a median coverage of fewer than five UMIs for this subset of probes were excluded from analysis.

Structural profiling was performed using the UMI count table (Supplementary Table 7). The count table was converted to a presence/absence table such that if a probe had more than one UMI for a given sample, it was accepted as present (that is, not deleted). Samples were clustered into subtelomeric structural profile groups based on this table using the hierarchical clustering algorithm AgglomerativeClustering of the Python module Scikit-learn (v.0.20)[56] using only the regions involved in the deletion events of the corresponding chromosome (position >1,372,615 for chromosome 8 and position > 2,806,319 for chromosome 13). Samples were grouped into their final subtelomeric structural profile based on visual inspection of the resulting clusters.

Initial variant calls were made using freebayes (v1.3.1) via MIPTools with the following options:--pooled-continuous--min-base-quality 1--min-alternate-fraction 0.01--min-alternate-count 2--haplotype-length -1--min-alternate-total 10--use-best-n-alleles 70--genotype-qualities. Variants were processed using MIPTools to filter for: variant quality >1, genotype quality >1, average alternate allele quality >15, minimum depth >2 UMIs; and make final genotype calls based on the major allele (within-sample allele frequency >0.5). In addition, the following variants were removed from the final call set: those that were observed as a major allele in less than two samples (singletons), not supported by more than two UMIs in at least three samples, present on multicopy genes, and indels. Variant calls were further filtered for missingness to avoid imputation in EHH calculations: samples missing calls for >50% of the variants were removed, variants missing calls in >50% of the samples were removed. Variants calls were converted to.map and.hap files (Supplementary Table 8) for use with the rehh package (v3.1.2) in R.

**Assessment of MIP calls using whole-genome sequencing.** We performed WGS on a subset of samples selected by convenience to assess the accuracy of MIP *pfhrp2/3* deletion calls. DNA extracted from samples with discordant RDT results were selected for *P. falciparum* selective whole-genome amplification (sWGA) and whole-genome sequencing exactly as described previously[31]. In brief, DNA was first subjected to two separate sWGA reactions using the Probe_10 primer set described by Oyola et al[57] and the JP9 primer set[31]. sWGA products were then pooled in equal volumes and acoustically sheared using a Covaris E220 instrument before to sequencing library preparation using Kappa Hyper library preps (Roche, catalogue no. KK8504). Indexed libraries were then pooled and sequenced on an Illumina HiSeq 4000 instrument using 150 bp, paired-end sequencing. Sequencing reads were deposited into NCBI's Sequence Read Archive (PRJNA742125).

**Published whole-genome sequencing data retrieval.** Fastq files from 25 Ethiopian samples included in the MalariaGEN genome variation project[19] and three laboratory strains (3D7, HB3 and DD2) from MalariaGEN genetic crosses project[58] were downloaded from the European Nucleotide Archive using fasterq-dump (v.2.10.8) and sample accession numbers on 19 September 2020 (Supplementary Table 9).

**WGS data analysis.** All fastq files were processed as follows. Adaptor and quality trimming was performed using Trimmomatic (v.0.39) with the recommended options (seed mismatches:2, palindrome clip threshold:30, simple clip threshold:10, minAdapterLength:2, keepBothReads LEADING:3 TRAILING:3 MINLEN:36). Trimmed fastq files were mapped to 3D7 reference genome (v3.0) concatenated to human genome (hg38, downloaded from the US National Institutes of Health National Center for Biotechnology and Information database on 2 December 2015, and available at https://www.ncbi.nlm.nih.gov/assembly/GCF_000001405.26/) to avoid incorrect mapping of reads originating from host DNA using bowtie2 (v.2.3.0) with the '--very-sensitive' option. Reads mapping to the parasite chromosomes were selected and optical duplicates were removed using the sambamba (v.0.7.1) view and markdup commands, respectively. Read coverage was calculated using samtools (v.1.9) depth command with options '-a -Q1 -d0', filtering reads with mapping quality of zero. Variants were called only for the regions of interest using freebayes (v.1.3.1) with the following options: '--use-best-n-alleles 70--pooled-continuous--min-alternate-fraction 0.01--min-alternate-count 2--min-alternate-total 10--genotype-qualities--haplotype-length -1--min-mapping-quality 15 -r region'. Regions of interest were from 300 kb centromeric to the deletions to chromosome ends (positions 1,074,000 to 1,472,805 and 2,505,000 to 2,925,236 for chromosomes 8 and 13, respectively).

Variants were filtered for: variant quality >20, genotype quality >15, average alternate allele quality >15, minimum depth >4 reads. In addition, the following variants were removed from the final call set: those that were never observed

as a major allele in any sample, not supported by more than ten reads in at least one sample, and indels. Final genotype calls were based on the major allele (within-sample allele frequency >0.5). Variant calls were further filtered for missingness to avoid imputation in EHH calculations: samples missing calls for >95% of the variants were removed, variants missing calls in >10% of the samples were removed.

Telomeric profiling of the published genomes was carried out by visual inspection of depth-of-coverage plots (Extended Data Figs. 8 and 9). Summary statistics were generated (Supplementary Table 10) using the Python pandas module (v.0.23).

**Statistical and population genetic analysis.** Data collected during the participant's study visit (clinical data and RDT results) were linked to laboratory results via the barcode number transcribed on DBS sent to the UNC and CDC laboratories. Samples in the dataset with missing or duplicate barcodes were arbitrated using original paper questionnaires by the EPHI data centre. An analysis dataset that included both PCR and field data was created including all samples that we could confidently merge by both barcode number and region label.

Statistical analysis was performed using R (v.3.6.0, R Core Team; www.R-project. org). All 95% CI values were generated using one-sample proportions test with Yates' continuity correction (R package binom.confint, v.1.1-1), and unweighted Cohen's kappa estimates were generated using the psych (v.1.8.1) and epiR (v.1.0-15) packages. Lilliefors test was used to test for normality, and *P* values calculated using chi-squared test for sex and the Kruskal–Wallis test for age and parasite density. ArcGIS (desktop v.10.5, ESRI) was utilized for mapping, with additional annotation performed using PowerPoint (v.16.31, Microsoft).

EHH statistics were calculated to evaluate the regions flanking the *pfhrp2* and *pfhrp3* genes for signatures of recent positive selection[36] using the rehh package (v.3.1.2)[59]. EHH statistics were calculated using the data2haplohh and calc_ehh functions, haplotype furcations were calculated using calc_furcation, and plots were generated using the package's plot function and annotated using Inkscape (v.0.92).

Complexity of infection (COI) for each sample was calculated using McCOILR (v.1.3.0, https://github.com/OJWatson/McCOILR), an Rcpp wrapper for THE REAL McCOIL[60] with the options maxCOI = 25, totalrun = 2,000, burnin = 500, M0 = 15, err_method = 3. The same variant set used in the EHH analysis was used for the COI calculations, except that variants whose within-sample allele frequency were between 0.05 and 0.95 were called heterozygote for COI analysis.

**Statistics and reproducibility.** Sample sizes were chosen based on the WHO protocol[30]. Sources of data and samples included in the study are outlined in Fig. 2. DBS samples were only collected from a subset of participants based on the WHO protocol. Molecular, immunological and sequencing assays were performed on random subsets selected by EPHI. Most analyses were limited to samples that could be matched unambiguously across datasets. For example, any DBS samples found to have identical participant IDs were excluded from analysis. Similarly, DBS labelled with a participant and region ID that did not match clinical data were excluded from most analyses. These accounted for a minority of participants. Discordances in participant IDs and DBS sample labels were resolved whenever possible.

Field staff were not blinded to malaria RDT results because they were used to inform clinical care according to national guidelines. *Pfhrp2/3* deletion calls using MIP sequencing were made by an investigator who was blinded to clinical data (including RDT results), HRP2 immunoassay results and *pfhrp2/3* deletion calls using PCR.

**Reporting Summary.** Further information on research design is available in the Nature Research Reporting Summary linked to this article.

## Data availability
MIP and genomic sequencing data are available through the Sequence Read Archive (PRJNA742125). De-identified datasets generated during the current study and used to make all figures are available as supplementary files or tables. Extended Data Figs. 6 and 7 were derived from genomic sequencing data made publicly available by MalariaGEN (https://www.malariagen.net/data, downloaded 19 September 2020). Extended Data Fig. 8 was derived from genomic sequencing data generated during this study and publicly available through MalariaGEN.

## Code availability
Code used during data analysis is available through GitHub at https://doi.org/10.5281/zenodo.5160363.

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

## Acknowledgements

We thank the research teams for conducting fieldwork and the participants for participating in the study. We also thank S. R. Meshnick posthumously for contributions to the laboratory analyses and data analysis, and Q. Cheng for advice during PCR assay optimization. The following reagents were obtained through BEI Resources, NIAID, NIH: genomic DNA from *P. falciparum* strain 3D7, MRA-102G, contributed by D. J. Carucci; *P. falciparum* strain HB3, MRA-155G, contributed by T. E. Wellems; *P. falciparum* strain Dd2, MRA-150G, contributed by D. Walliker. We also acknowledge MSF Holland for supporting the field study in the Gambella region. The findings and conclusions in this report are those of the authors and do not necessarily represent the official position of the Centers for Disease Control and Prevention. This work was funded by the Global Fund to Fight AIDS, Tuberculosis, and Malaria through the Ministry of Health-Ethiopia (EPHI5405 to S.M.F.) and by the Bill and Melinda Gates Foundation through the World Health Organization (OPP1209843 to J.C., J.B.P.). It was also partially supported by MSF Holland, which supported fieldwork in the Gambella region, the Doris Duke Charitable Foundation (J.B.P.), the American Society for Tropical Medicine and Hygiene-Burroughs Wellcome Foundation (J.B.P.) and the US National Institutes of Health (R01AI132547 to J.J.J., J.A.B., O.A. and J.B.P.; K24AI134990 to J.J.J.). Under the grant conditions of the Bill and Melinda Gates Foundation, a Creative Commons Attribution 4.0 Generic License has already been assigned to the Author Accepted Manuscript version that might arise from this submission.

## Author contributions

S.M.F., J.C. and J.B.P. conceived the study. S.M.F., H. Mohammed and B.G.B. led the fieldwork, supervised by H. Mamo, H.S., B.P. and E.A. O.A., C.H., M.D. and E.R. performed laboratory assays and experiments. E.N.R., O.A., C.K., J.J.J., J.A.B., E.R. and J.B.P. analysed the laboratory data. E.N.R. and O.A. produced the tables and figures. S.M.F. and E.N.R. wrote the first draft with assistance from O.A. and J.B.P. All authors critically reviewed and approved the final manuscript.

## Competing interests

J.B.P. reports research support from Gilead Sciences and non-financial support from Abbott Diagnostics for studies of viral hepatitis, and honorarium from Virology Education for medical education teaching on COVID-19, all outside the scope of the current work. S.M.F. reports research support from Access Bio for a separate study of malaria RDTs, outside the current work. All other authors declare no competing interests.

## Additional information

**Extended data** is available for this paper at https://doi.org/10.1038/s41564-021-00962-4.

**Correspondence and requests for materials** should be addressed to Sindew M. Feleke, Jane Cunningham or Jonathan B. Parr.

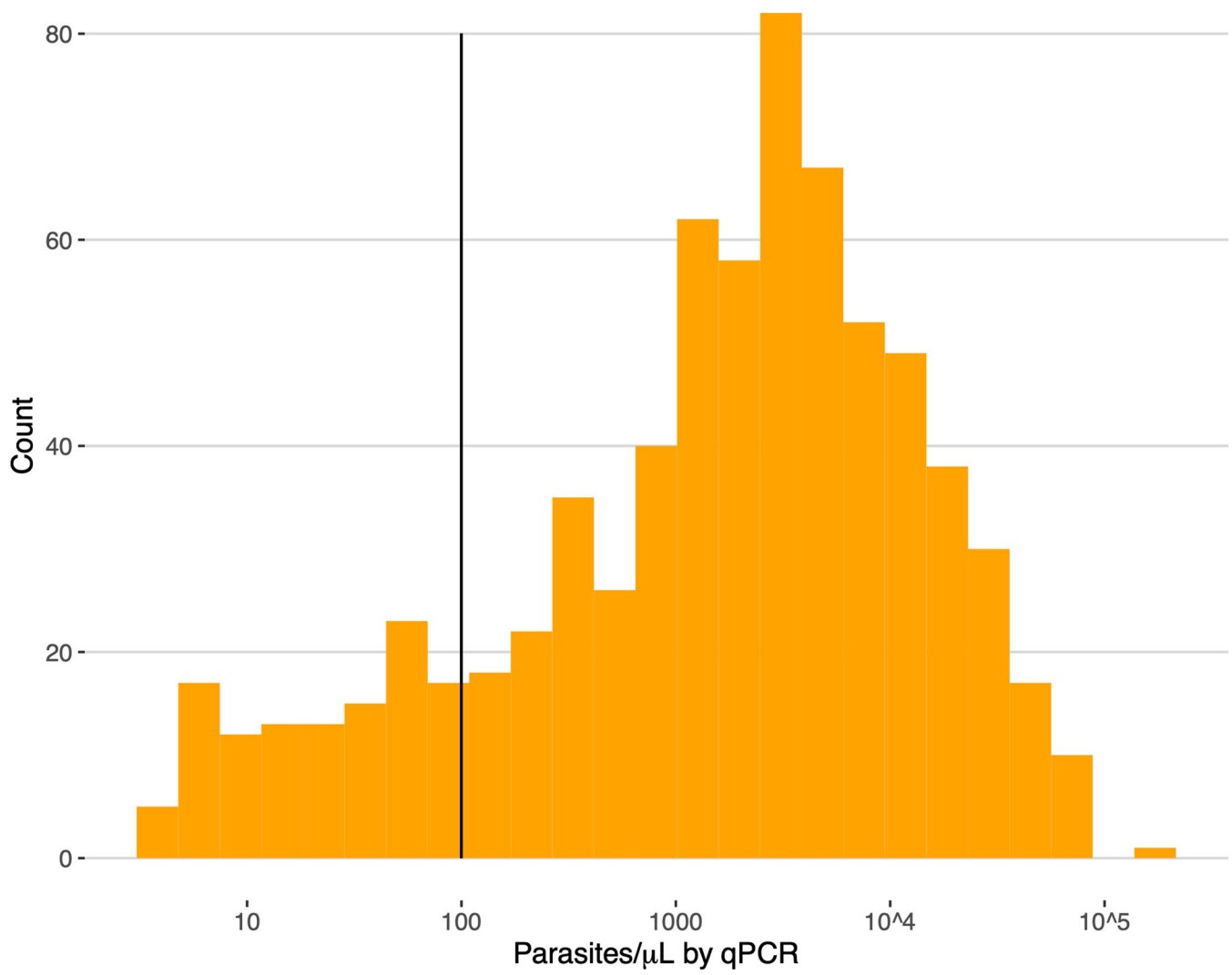

**Extended Data Fig. 1 | Parasite density distribution.** *Pfldh* quantitative PCR (qPCR) results used to assess parasite density and determine which samples were eligible for *pfhrp2/3* deletion genotyping using a series of PCR assays. *Pfhrp2/3* deletions were only called in 610 samples with >100 parasites/μL (solid line).

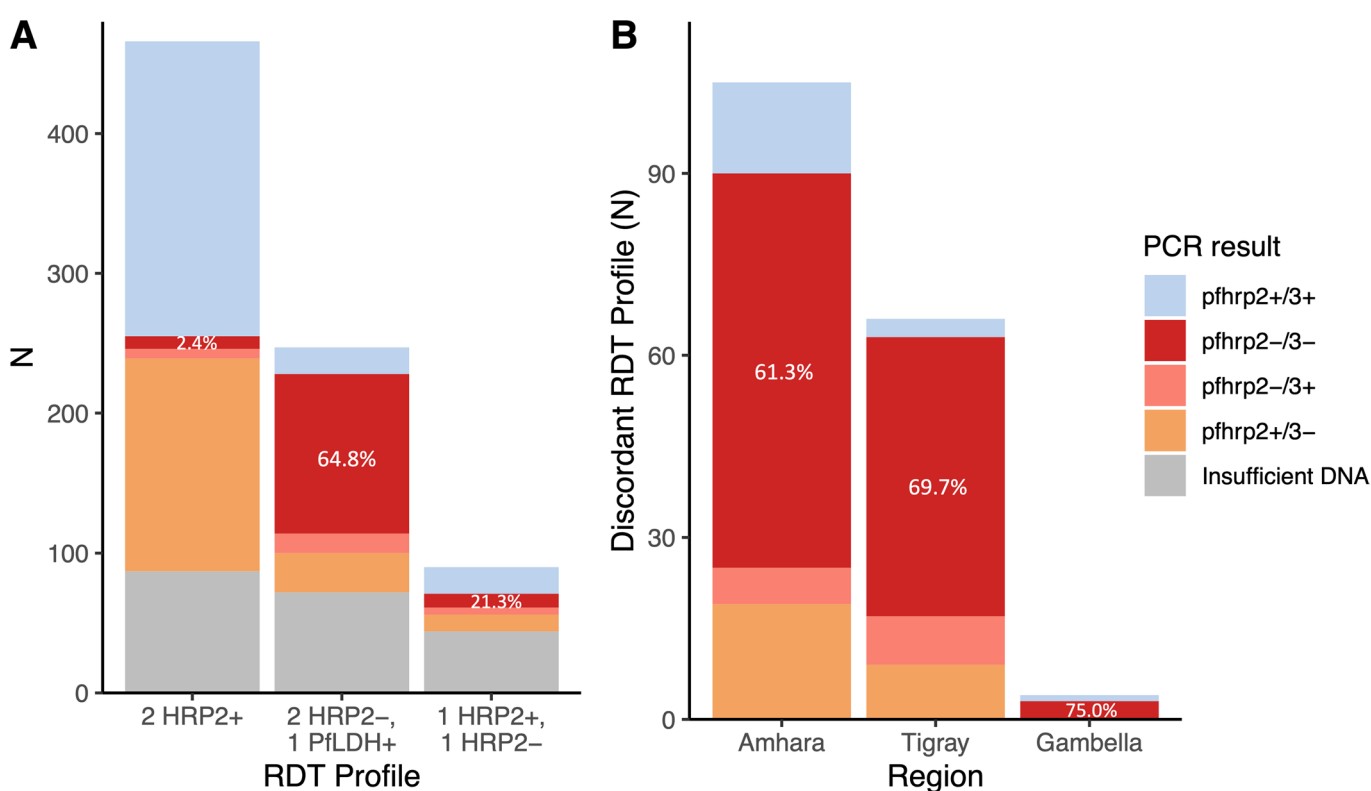

**Extended Data Fig. 2 | *Pfhrp2* and *pfhrp3* PCR results by RDT profile.** A) Concordance between RDT profile and PCR *pfhrp2/3* result for 602 *P. falciparum* samples with >100 parasites/µL. **b**) *Pfhrp2/3* PCR results for participants with the discordant RDT profile and sufficient DNA for molecular analysis (n = 176), by study region.

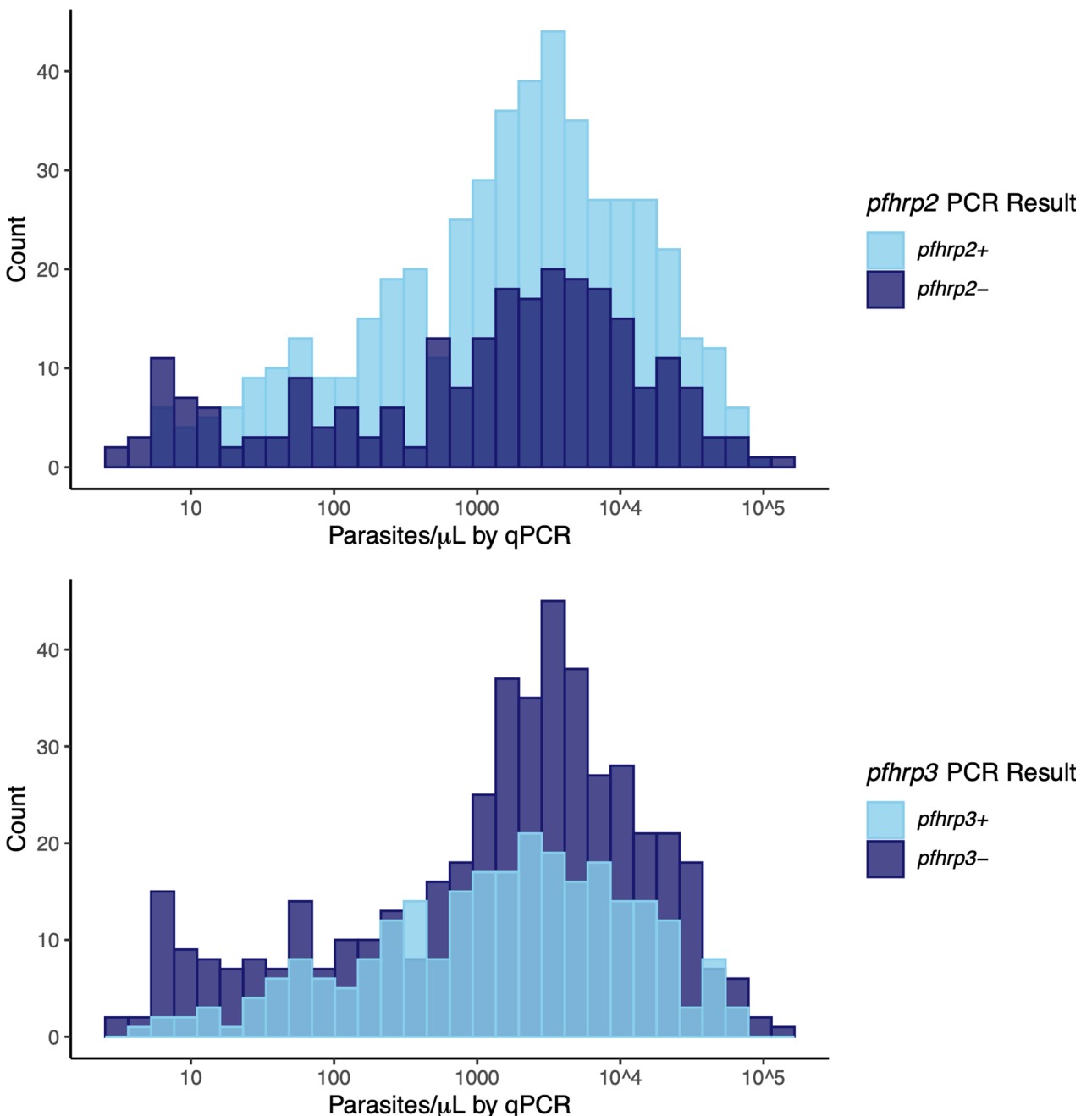

**Extended Data Fig. 3 | *Pfhrp2/3* PCR results versus qPCR parasite density.** Comparison of *pfhrp2* (top) and *pfhrp3* (bottom) PCR results versus *pfldh* qPCR parasite densities. Note that final *pfhrp2/3* PCR genotyping calls were only made in samples with qPCR parasite densities >100 parasites/μL and a positive *P. falciparum* beta-tubulin real-time PCR result to avoid misclassification of deletions in the setting of low DNA target concentrations.

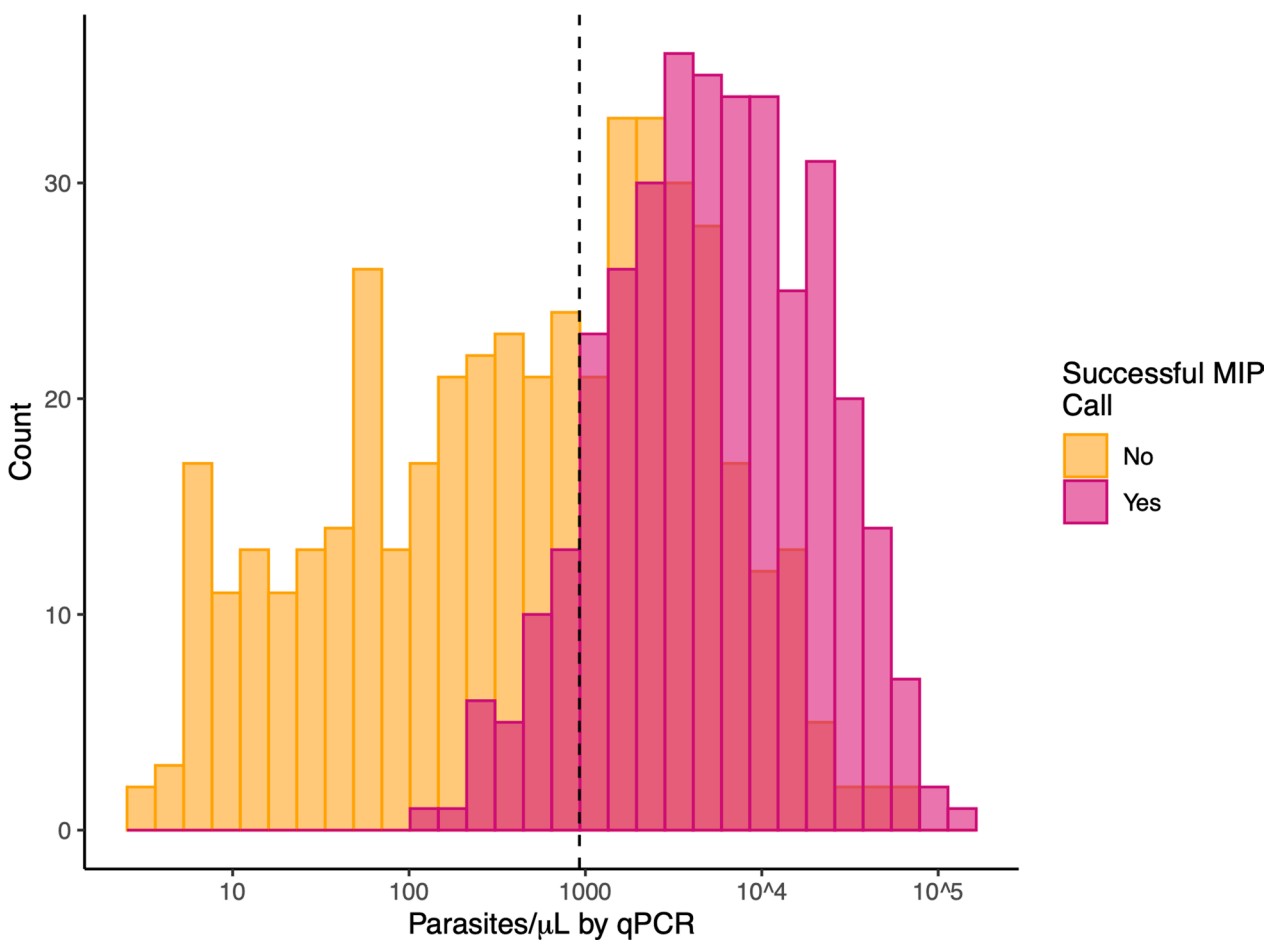

**Extended Data Fig. 4 | Successful MIP deletion calls versus qPCR parasite density.** Comparison of MIP call results and qPCR parasite densities suggests a project-specific threshold for MIP calling of approximately 925 p/μL of whole blood (dashed line).

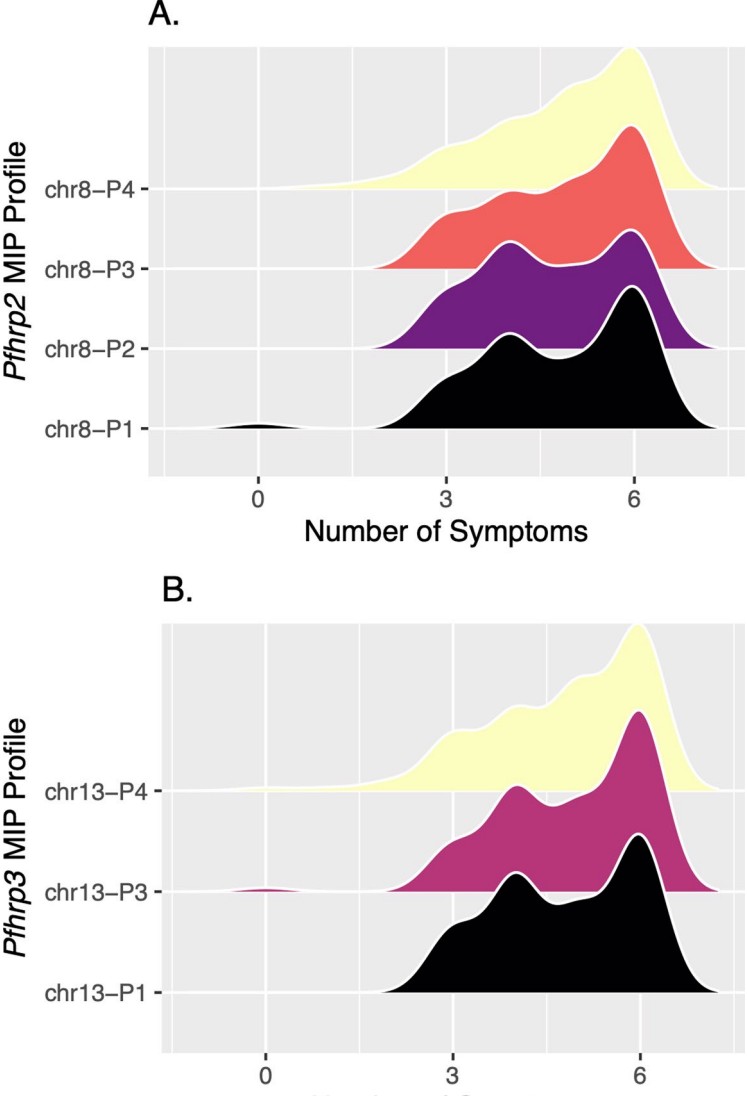

**Extended Data Fig. 5 | Disease severity by subtelomeric structural profile.** Smoothed distribution of disease severity for each of the broader deletion breakpoint haplotypes along chromosomes 8 (A) and 13 (B) identified by MIP genomic enrichment among MIP samples with matching clinical data (n = 338). The total number of symptoms with which participants presented was used to estimate disease severity, with all participants evaluated for: fever, headache, joint pain, feeling cold, nausea, and loss of appetite. Profile chr13-P2 was excluded due to its small sample size (n=1).

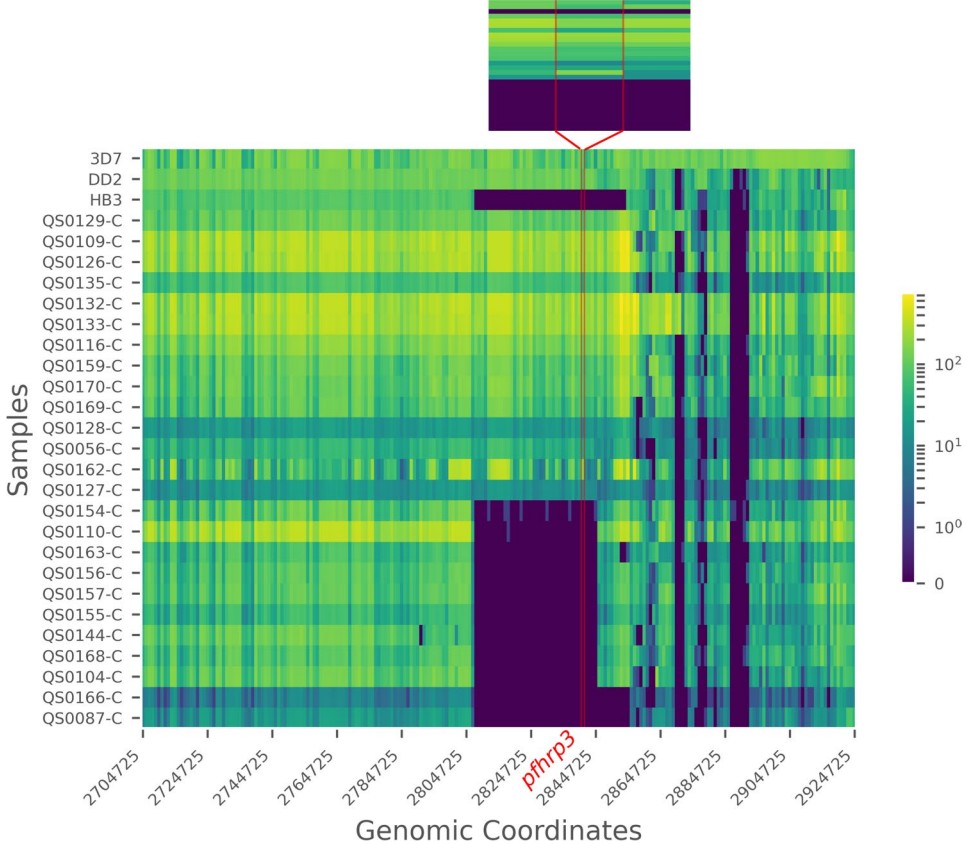

**Extended Data Fig. 6 | Chromosome 13 telomeric-end coverage (aligned reads/locus) plot of WGS control strains and 25 published Ethiopian genomes from 2013–2015 (MalariaGEN).** The location of *pfhrp3* is indicated by vertical red lines. Inset showing *pfhrp3* and 1 kilobase flanking genomic region. Large subtelomeric deletions containing *pfhrp3* are apparent in the laboratory strain HB3, as well as 11 samples from Ethiopia.

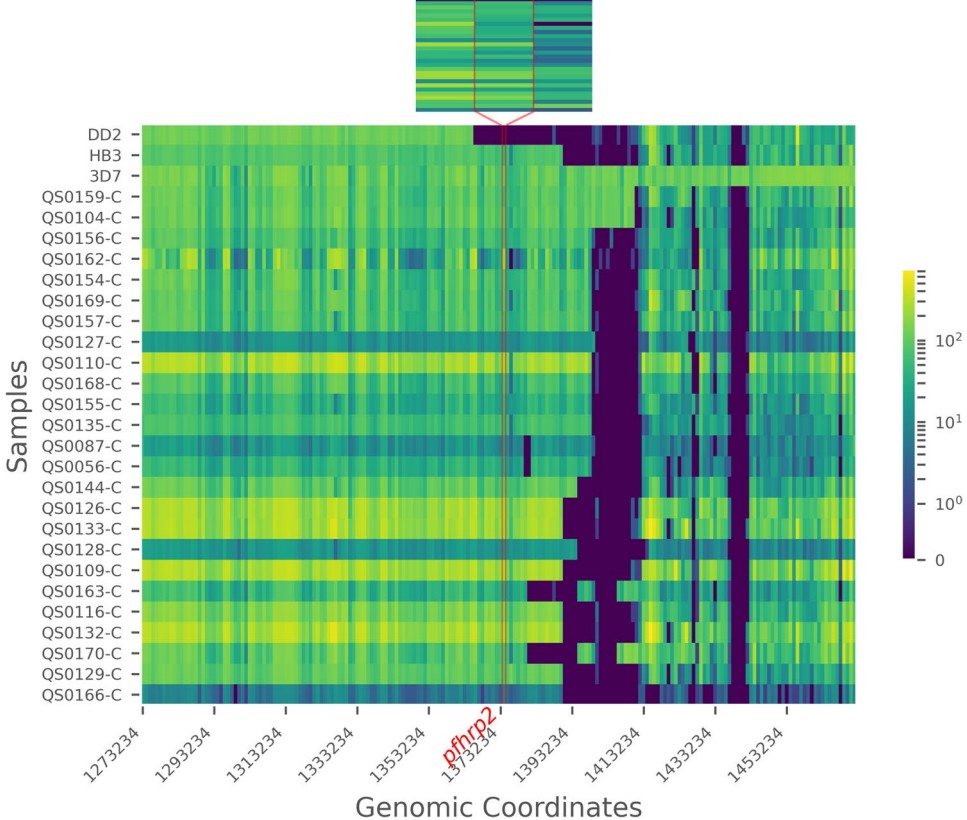

**Extended Data Fig. 7 | Chromosome 8 telomeric-end coverage (aligned reads/locus) plot of WGS control strains and 25 published Ethiopian genomes from 2013–2015 (MalariaGEN).** The location of *pfhrp2* is indicated by vertical red lines. Inset showing *pfhrp2* and 1 kilobase flanking genomic region. Large subtelomeric deletions compared to the reference strain 3D7 are apparent in most samples. None of the deletions involve *pfhrp2* except the DD2 strain.

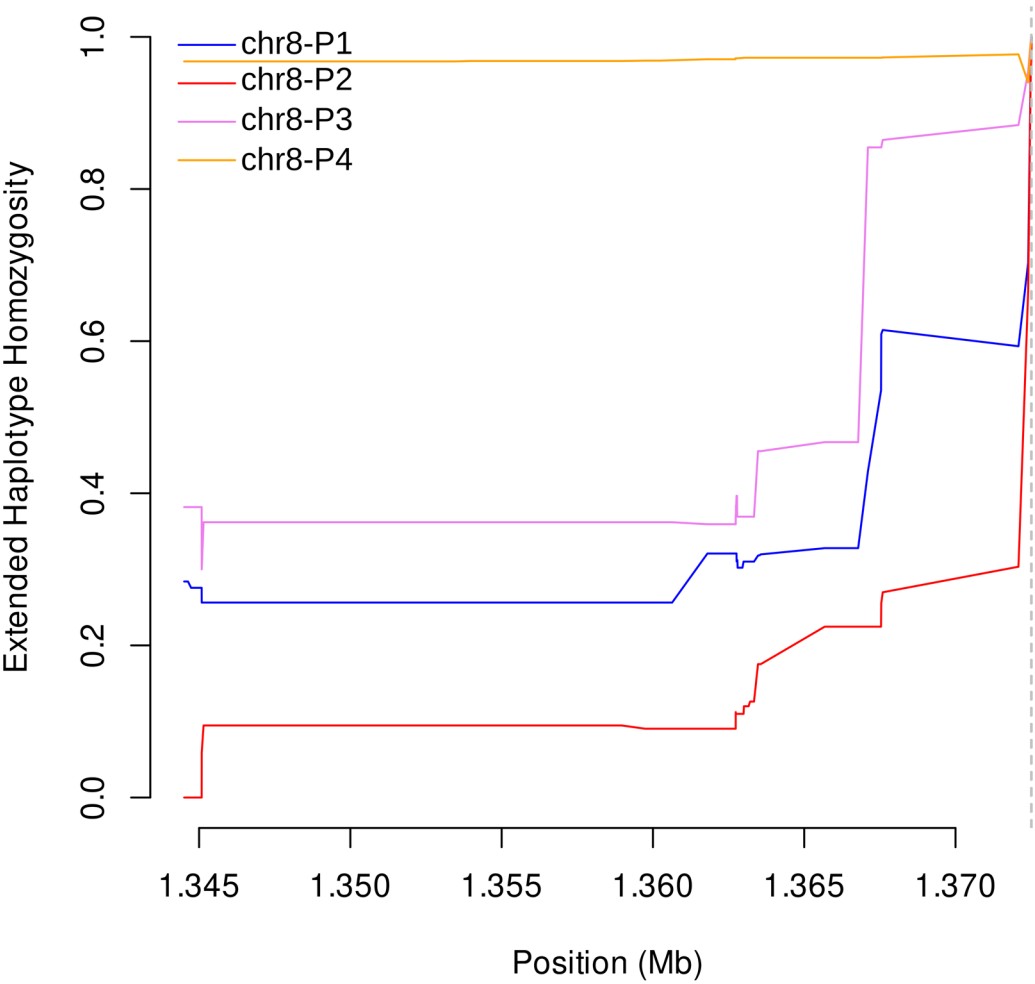

**Extended Data Fig. 8 | Extended haplotype homozygosity centromeric to chromosome 8 subtelomeric structural profiles P1-P4 using MIP data.** The only *pfhrp2*-deleted profile (chr8-P4) showed sustained EHH, whereas EHH quickly broke down for the *pfhrp2*-intact profiles (P1-P3). A vertical dashed line on the right marks the centromeric end of the profile 4 (*pfhrp2*) deletion.

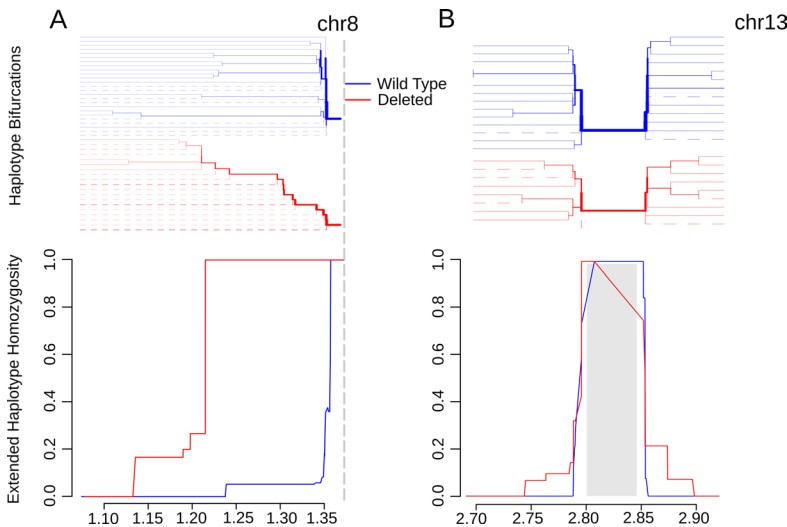

**Extended Data Fig. 9 | Extended haplotype homozygosity (bottom) and the bifurcation diagrams showing haplotype branching (top) centromeric to the** *pfhrp2* **(A) and surrounding** *pfhrp3* **(B) deletions based on WGS data.** Vertical dashed line indicating the centromeric end of the chromosome 8 deletion (**a**). Gray box demarcating the chromosome 13 deletion (**b**). Abbreviations: Mb, mega-base.

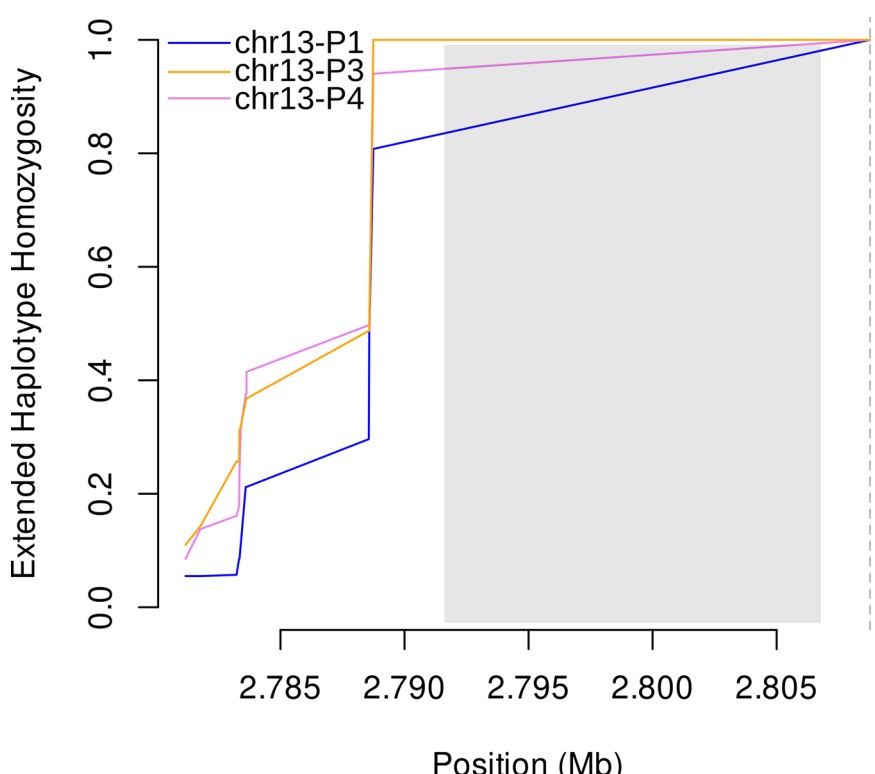

**Extended Data Fig. 10 | Extended haplotype homozygosity centromeric to chromosome 13 subtelomeric structural profiles P1, P3 and P4 using MIP data.** Chr13-P2 profile was observed in only one sample and not included in the haplotype analysis. EHH quickly brown down for all profiles (P1: *pfhrp3*-intact, P3-P4: *pfhrp3*-deleted). A vertical dashed line on the right marks the centromeric end of the P3 and P4 deletions. No variants in the duplicated segment (gray box) were used in the EHH analysis.

# nature research

| | |
|---|---|

# Reporting Summary

Nature Research wishes to improve the reproducibility of the work that we publish. This form provides structure for consistency and transparency in reporting. For further information on Nature Research policies, see our Editorial Policies and the Editorial Policy Checklist.

## Statistics

For all statistical analyses, confirm that the following items are present in the figure legend, table legend, main text, or Methods section.

| n/a | Confirmed | |
|---|---|---|
| ☐ | ☒ | The exact sample size (*n*) for each experimental group/condition, given as a discrete number and unit of measurement |
| ☐ | ☒ | A statement on whether measurements were taken from distinct samples or whether the same sample was measured repeatedly |
| ☐ | ☒ | The statistical test(s) used AND whether they are one- or two-sided *Only common tests should be described solely by name; describe more complex techniques in the Methods section.* |
| ☐ | ☒ | A description of all covariates tested |
| ☐ | ☒ | A description of any assumptions or corrections, such as tests of normality and adjustment for multiple comparisons |
| ☐ | ☒ | A full description of the statistical parameters including central tendency (e.g. means) or other basic estimates (e.g. regression coefficient) AND variation (e.g. standard deviation) or associated estimates of uncertainty (e.g. confidence intervals) |
| ☐ | ☒ | For null hypothesis testing, the test statistic (e.g. *F*, *t*, *r*) with confidence intervals, effect sizes, degrees of freedom and *P* value noted *Give P values as exact values whenever suitable.* |
| ☒ | ☐ | For Bayesian analysis, information on the choice of priors and Markov chain Monte Carlo settings |
| ☒ | ☐ | For hierarchical and complex designs, identification of the appropriate level for tests and full reporting of outcomes |
| ☒ | ☐ | Estimates of effect sizes (e.g. Cohen's *d*, Pearson's *r*), indicating how they were calculated |

*Our web collection on statistics for biologists contains articles on many of the points above.*

## Software and code

Policy information about availability of computer code

| Data collection | Data was collected in the field using paper forms and double entered into Epi Info (v3.2). Discrepancies were cross-checked against the hard-copy paper forms and resolved by consensus. |
|---|---|
| Data analysis | Custom code used during data analysis is available through GitHub at https://doi.org/10.5281/zenodo.5160363. Statistical analysis was performed using R (version 3.6.0, R Core Team, Vienna, Austria, 2019; www.R-project.org). Prevalence estimates were generated using the asbio package (v1.5-5). 95% confidence intervals were calculated using the binom.confint package (v1.1-1), and unweighted Cohen's kappa estimates calcualted using the psych (v1.8.1) and epiR (v1.0-15) packages. ArcGIS (Desktop Version 10.5, ESRI, Redlands, CA, 2016) was utilized for mapping, with additional annotation performed using PowerPoint (version 16.31, Microsoft, Redmond, WA, 2019).<br><br>MIP design and sequencing fastq file processing were performed using MIPTools (v0.19.12.13), which uses the MIPWrangler algorithm (v1.2.0), bwa (v0.7.17), and freebayes (v1.3.1). Structural profile groups were assigned using the hierarchical clustering algorithm AgglomerativeClustering of the Python module Scikit-learn (v0.20).<br><br>Genome sequencing fastq files were processed and analysed using Trimmomatic (v.0.39), bowtie2 (v2.3.0), sambamba (v0.7.1), samtools (v1.9), and freebayes (v1.3.1). Publicly available parasite genomes from Ethiopia were downloaded using fasterq-dump (v2.10.8).<br><br>Extended haplotype homozygosity (EHH) statistics were calculated using the rehh package (version 3.1.2); plots were annotated using Inkscape (version 0.92). Complexity of infection (COI) was calculated using McCOILR (v1.3.0, https://github.com/OJWatson/McCOILR), an Rcpp wrapper for THE REAL McCOIL. |

For manuscripts utilizing custom algorithms or software that are central to the research but not yet described in published literature, software must be made available to editors and reviewers. We strongly encourage code deposition in a community repository (e.g. GitHub). See the Nature Research guidelines for submitting code & software for further information.

## Data

Policy information about availability of data

All manuscripts must include a data availability statement. This statement should provide the following information, where applicable:

- Accession codes, unique identifiers, or web links for publicly available datasets
- A list of figures that have associated raw data
- A description of any restrictions on data availability

MIP and genomic sequencing data is available through the Sequence Read Archive (PRJNA742125). De-identified datasets generated during the current study and used to make all figures are available as supplementary files or tables.

Extended Data Figure 6-7 were derived from genomic sequencing data made publicly available by MalariaGEN (https://www.malariagen.net/data, downloaded Sep 19 2020). Extended Data Figure 8 was derived from genomic sequencing data generated during this study and publicly available through MalariaGEN.

The hg38 human genome used during whole-genome sequencing analysis was downloaded from the US National Institutes of Health National Center for Biotechnology and Information database on December 2, 2015 (available at https://www.ncbi.nlm.nih.gov/assembly/GCF_000001405.26/).

# Field-specific reporting

Please select the one below that is the best fit for your research. If you are not sure, read the appropriate sections before making your selection.

☒ Life sciences          ☐ Behavioural & social sciences          ☐ Ecological, evolutionary & environmental sciences

For a reference copy of the document with all sections, see nature.com/documents/nr-reporting-summary-flat.pdf

# Life sciences study design

All studies must disclose on these points even when the disclosure is negative.

| | |
|---|---|
| Sample size | Sample sizes were derived from the WHO "Template protocols to support surveillance and research for pfhrp2/pfhrp3 gene deletions," available at https://www.who.int/malaria/publications/atoz/hrp2-deletion-protocol/en/. |
| Data exclusions | Sources of data and samples included in the study are outlined in Figure 2. Dried blood spot samples were only collected from a subset of subjects based on WHO protocols. Molecular, immunological, and sequencing assays were performed on random subsets selected by EPHI. As outlined in the Methods, most analyses were limited to samples that could be matched unambiguously across datasets. For example, any DBS samples found to have identical participant IDs were excluded from analysis. Similarly, DBS labeled with a participant and region ID that did not match clinical data were excluded from most analyses. These accounted for a minority of subjects. Discordances in participant IDs and DBS sample labels were resolved whenever possible. |
| Replication | All PCR assays were performed in duplicate. Deletion calls made by PCR were limited to samples with >100 parasites/μL, with negative pfhrp2 or pfhrp3 bands in both replicates, and positive by a final confirmatory real-time PCR assay as described in the Methods.

Comparison of whole-genome sequencing and MIP calls was undertaken for 14 samples as outlined in the Results. Sequencing (MIP and WGS) was performed across multiple flow cells.

To increase confidence in pfhrp2/3 deletion calls, multiple confirmatory methods were employed, including PCR, MIP sequencing, WGS, and an HRP2 immunoassay.

Results were compared across platforms, and concordance/discordance between methods included in the Results. While most calls were concordant, we did observe samples with discordance results across different assays. This was not unexpected because the assay targets are different in some cases (ex: HRP2 immunoassay detects HRP2 or HRP3 antigen that can linger after clearance of parasitemia, whereas the molecular methods detect parasite DNA that clears rapidly after resolution of infection). However, we cannot exclude the possibility that some discordance may have been introduced by ambiguous sample labeling and/or processing during the conduct of the field work. We overcame this by restricting analyses to samples with complete meta-data and no ambiguity when merged with molecular and/or antigen data (see Figure 2), and by employing a conservative approach to prevalence estimates as described in the Results, Methods, and Discussion. |
| Randomization | As outlined in the WHO protocol, any subject presenting to study health facilities with symptoms of malaria was eligible for enrollment. Randomization was not performed. We did not undertake detailed analyses of covariates, except as shown in Supplementary Table 1 in which we stratified by pfhrp2/3 status. |
| Blinding | Field staff were not blinded to malaria RDT results because they were used to inform clinical care according to national guidelines.

Pfhrp2/3 deletion calls using MIP sequencing were made by an investigator who was blinded to clinical data (including RDT results), HRP2 immunoassay results, and pfhrp2/3 deletion calls using PCR. |

# Reporting for specific materials, systems and methods

We require information from authors about some types of materials, experimental systems and methods used in many studies. Here, indicate whether each material, system or method listed is relevant to your study. If you are not sure if a list item applies to your research, read the appropriate section before selecting a response.

## Materials & experimental systems

| n/a | Involved in the study |
|-----|-----------------------|
| ☒ | ☐ Antibodies |
| ☒ | ☐ Eukaryotic cell lines |
| ☒ | ☐ Palaeontology and archaeology |
| ☒ | ☐ Animals and other organisms |
| ☐ | ☒ Human research participants |
| ☒ | ☐ Clinical data |
| ☒ | ☐ Dual use research of concern |

## Methods

| n/a | Involved in the study |
|-----|-----------------------|
| ☒ | ☐ ChIP-seq |
| ☒ | ☐ Flow cytometry |
| ☒ | ☐ MRI-based neuroimaging |

## Human research participants

Policy information about studies involving human research participants

| | |
|---|---|
| Population characteristics | Subjects of all ages and genders who presented to health facilities with symptoms of malaria were eligible for enrollment. All subjects provided informed consent. Participants were not compensated. See Methods for details.<br><br>Among 12,572 study subjects, the median age was 19 years (interquartile range 8-30). 5,555 (44.2%) were female. See Table 1 for details. |
| Recruitment | This was a cross-sectional survey conducted at 108 health facilities located in eleven districts within three regions of Ethiopia near its borders with Eritrea, Sudan, and South Sudan. Subjects were enrolled as noted above, but DBS samples for molecular and/or immunological analysis were only collected from a subset. The study was designed to collect DBS from any subject with a 'discordant RDT result' suggestive of infection by pfhrp2/3-deleted P. falciparum (HRP2 bands negative on two distinct RDTs, Pf-LDH band positive) and from 10-20% of subjects with positive HRP2 bands. The focus on discordant RDT results is an intentional component of the WHO protocol, included to allow real-time, efficient signaling to malaria control programs.<br><br>This was a pragmatic survey conducted as part of routine malaria care at government health facilities. As such, DBS were not collected/available from all subjects with discordant RDT results (see Results). In addition, not all samples underwent pfhrp2/3 deletion PCR genotyping (to avoid the risk of misclassification due to low P. falciparum DNA concentrations), HRP2 immunoassay, or sequencing. Among those sequenced, only a subset had sufficient UMI depth of coverage to be included in analysis.<br><br>These features and the study design could introduce selection bias when estimating prevalence. For example, a disproportionate number of subjects with a 'discordant RDT result' had DBS sent for molecular and antigen testing, which could lead to over-estimates of pfhrp2/3 deletion prevalence. We accounted for this by estimating the frequency of falciparum malaria cases with false-negative RDT results due to pfhrp2/3 deletions using RDT data from the highest-level dataset (12,572 participants - see Figure 2). This dataset was derived from all enrolled subjects and therefore felt to be most representative of people presenting for routine malaria care.<br><br>Figure 2 displays how subjects and samples were included in analyses, and denominators are included throughout the Results to avoid ambiguity. |
| Ethics oversight | Ethical approval was obtained from the Ethiopia Public Health Institute (EPHI) Institutional Review Board (IRB; protocol EPHI-IRB-033-2017) and WHO Research Ethics Review Committee (protocol: ERC.0003174 001). |

Note that full information on the approval of the study protocol must also be provided in the manuscript.

