## [Peer Review File. · Nature Microbiology]

Peer Review Information

Journal: Nature Microbiology

Manuscript Title: Plasmodium falciparum is evolving to escape malaria rapid diagnostic tests in Ethiopia

Corresponding author name(s): Jonathan Parr

Reviewer Comments & Decisions:

Decision Letter, initial version:
--

Dear Dr Parr,

Thank you for your patience while your manuscript "Plasmodium falciparum histidine-rich protein 2 deletion is under recent positive selection in Ethiopia and threatens malaria diagnostic strategies" was under peer-review at Nature Microbiology. It has now been seen by the same 2 referees who provided input on the first version of this manuscript. You will see from their comments below that reviewer 1 offers guidance on further improvement to your paper, most of which are minor and involve clarifications, more detail in the methods and moving Supplementary items to the main text. We are very interested in the possibility of publishing your study in Nature Microbiology, and now encourage you to revise your manuscript using the comments from reviewer 1 as guidance.

If you have not done so already please begin to revise your manuscript so that it conforms to our Article format instructions at <http://www.nature.com/nmicrobiol/info/final-submission/>

The usual length limit for a Nature Microbiology Article is 6-8 display items (figures or tables) and 3,000 words. We have some flexibility, and can allow a revised manuscript at 3,500 words, but please consider this a firm upper limit.

We strongly support public availability of data. Please place the data used in your paper into a public data repository, if one exists, or alternatively, present the data as Source Data or Supplementary Information. If data can only be shared on request, please explain why in your Data Availability Statement, and also in the correspondence with your editor. For some data types, deposition in a public repository is mandatory - more information on our data deposition policies and available

repositories can be found at <https://www.nature.com/nature-research/editorial-policies/reporting-standards#availability-of-data>.

Please include a data availability statement as a separate section after Methods but before references, under the heading "Data Availability". This section should inform readers about the availability of the data used to support the conclusions of your study. This information includes accession codes to public repositories (data banks for protein, DNA or RNA sequences, microarray, proteomics data etc...), references to source data published alongside the paper, unique identifiers such as URLs to data repository entries, or data set DOIs, and any other statement about data availability. At a minimum, you should include the following statement: "The data that support the findings of this study are available from the corresponding author upon request", mentioning any restrictions on availability. If DOIs are provided, we also strongly encourage including these in the Reference list (authors, title, publisher (repository name), identifier, year). For more guidance on how to write this section please see:

<http://www.nature.com/authors/policies/data/data-availability-statements-data-citations.pdf>

To improve the accessibility of your paper to readers from other research areas, please pay particular attention to the wording of the paper's opening bold paragraph, which serves both as an introduction and as a brief, non-technical summary in about 150 words. If, however, you require one or two extra sentences to explain your work clearly, please include them even if the paragraph is over-length as a result. The opening paragraph should not contain references. Because scientists from other sub-disciplines will be interested in your results and their implications, it is important to explain essential but specialised terms concisely. We suggest you show your summary paragraph to colleagues in other fields to uncover any problematic concepts.

If your paper is accepted for publication, we will edit your display items electronically so they conform to our house style and will reproduce clearly in print. If necessary, we will re-size figures to fit single or double column width. If your figures contain several parts, the parts should form a neat rectangle when assembled. Choosing the right electronic format at this stage will speed up the processing of your paper and give the best possible results in print. We would like the figures to be supplied as vector files - EPS, PDF, AI or postscript (PS) file formats (not raster or bitmap files), preferably generated with vector-graphics software (Adobe Illustrator for example). Please try to ensure that all figures are non-flattened and fully editable. All images should be at least 300 dpi resolution (when figures are scaled to approximately the size that they are to be printed at) and in RGB colour format. Please do not submit Jpeg or flattened TIFF files. Please see also 'Guidelines for Electronic Submission of Figures' at the end of this letter for further detail.

Figure legends must provide a brief description of the figure and the symbols used, within 350 words, including definitions of any error bars employed in the figures.

- that unprocessed scans are clearly labelled and match the gels and western blots presented in figures.
- that control panels for gels and western blots are appropriately described as loading on sample processing controls

-- all images in the paper are checked for duplication of panels and for splicing of gel lanes.

Please include a statement before the acknowledgements naming the author to whom correspondence and requests for materials should be addressed.

Finally, we require authors to include a statement of their individual contributions to the paper -- such as experimental work, project planning, data analysis, etc. -- immediately after the acknowledgements. The statement should be short, and refer to authors by their initials. For details please see the Authorship section of our joint Editorial policies at http://www.nature.com/authors/editorial_policies/authorship.html

- * include a point-by-point response to any editorial suggestions and to our referees. Please include your response to the editorial suggestions in your cover letter, and please upload your response to the referees as a separate document.

- * ensure it complies with our format requirements for Articles as set out in our guide to authors at www.nature.com/nmicrobiol/info/gta/

- * state in a cover note the length of the text, methods and legends; the number of references; number and estimated final size of figures and tables

- * resubmit electronically if possible using the link below to access your home page:

{REDACTED}

- *This url links to your confidential homepage and associated information about manuscripts you may have submitted or be reviewing for us. If you wish to forward this e-mail to co-authors, please delete this link to your homepage first.

Please ensure that all correspondence is marked with your Nature Microbiology reference number in the subject line.

Nature Microbiology is committed to improving transparency in authorship. As part of our efforts in this direction, we are now requesting that all authors identified as 'corresponding author' on published papers create and link their Open Researcher and Contributor Identifier (ORCID) with their account on the Manuscript Tracking System (MTS), prior to acceptance. This applies to primary research papers only. ORCID helps the scientific community achieve unambiguous attribution of all scholarly contributions. You can create and link your ORCID from the home page of the MTS by clicking on 'Modify my Springer Nature account'. For more information please visit www.springernature.com/orcid.

We hope to receive your revised paper within three weeks. If you cannot send it within this time, please let us know.

{REDACTED}

Reviewers Comments:

Reviewer #1 (Remarks to the Author):

This article by Feleke et al. presents a comprehensive assessment of two rapid diagnostic tests (RDTs) employed to detect *Plasmodium falciparum* (Pf) malaria in Ethiopia. Both tests rely on the detection of Pf HRP2 antigen in whole blood. CareStart includes a test for *P. vivax* LDH, whereas SD Bioline includes a test for Pf LDH. The authors report the compelling finding that nearly 10% of all Pf infections sampled in Ethiopia (in their three sites) are HRP2 negative and thus would escape RDT detection. This finding was made through a powerful combination of RDT data from the two tests, PCR, molecular inversion probes and some whole-genome data. These state of the art genetic and genomic techniques, applied to a large data set of 2,714 Pf-positive samples (from a larger screen of 12,572 participants) yielded evidence that parasites with HRP2 deletions are closely related at the population level and that HRP3 deletions, which are more prevalent, occurred earlier and are more heterogeneous. The HRP2 data suggest a selective sweep arising from parasite escape of RDT screening prior to treatment. These data highlight a critical weakness in using these RDTs to detect and treat Pf malaria.

In addition to reviewing this submission, I have read the critiques from the prior submission to Nature Medicine, and feel that the authors responded very well. Overall, I find this an important study rich in data that will be of substantial interest to the malaria control and research community and other investigators studying antimicrobial diagnostics and population structures associated with intensive pathogen screening prior to treatment. I nonetheless have a few additional queries and comments:

1. Greater clarity should be provided about the way these tests work. The CareStart literature that I saw (<https://www.apacor.com/wp-content/uploads/2019/01/APA059-v8-1630-16305-CareStart-Malaria-Test-Procedure-updated-1.pdf>) indicates that one band is specific to Pf HRP2 and the other will appear positive if there is detectable LDH antigen from any of the Plasmodium species. Therefore a LDH+ band combined with a HRP2- band could indicate either a non-Pf species or Pf that lacks the *hrp2* gene. The manuscript states that the CareStart is Pf HRP2 / Pv-specific LDH. The authors should clarify this apparent discrepancy. If the LDH is indeed pan specific that would change a lot of their interpretation. For the SD Bioline assay I found it surprisingly difficult to obtain information about how the assay works. Some versions on the internet show that there is only one band for Pf, which would conflate HRP2 and LDH. The manuscript indicates that the SD Bioline recognizes both Pf HRP2 and Pf-specific LDH. The authors need to clarify this. Also, they should point out that each assay has a control band. Ideally, it would help to have an additional supplementary figure that shows both types of RDTs with a description of how they work and how different banding patterns should be interpreted. The authors should also include links in their additional supplementary figure (there may be several

versions of each test, leading to the apparent discrepancy above). The issue of species specific versus pan-Plasmodium is a critical one to address.

2. On a similar note, I think it would be very helpful to readers to have Supplementary Table 11 moved to a main Table as that shows the differences between the two assays and also documents the 332 (~9.7%) of samples that were Pf LDH+, HRP2- and Pv-LDH-. That table explicitly states that the CareStart has a Pv-specific LDH, which the authors should clarify in response to comment 1.

3. It would be helpful to know the prevalence of *P. vivax* in their cohort, even though I recognize this is not central to their study. They describe *P. vivax* data in the supplement (lines 863-878) and it would help to give a percentage (or estimate?) of the number of *P. vivax* samples as defined by their CareStart RDT. Were there any *P. vivax* mono-infections or were they always with Pf? The authors should also expand on any other evidence for *P. vivax* that they obtained using other molecular techniques (PCR? WGS?). If the Editors allow more space, it would be useful to include this section in the main manuscript.

4. The authors make the worrying statement (lines 165-167) that sufficient cross-reactive HRP3 can trigger a positive HRP2 band. They then exclude hrp2- hrp3+ samples with sufficient cross-reactive HRP3. How do they determine which samples have sufficient cross-reactive HRP3? This seems like a difficult factor to correct. Any quantitative data in this regard or more explanation would be very helpful.

5. On page 386 the authors state that the prevalence of hrp2/hrp3-deleted parasites appears to have been stable in neighboring Eritrea despite removal of HRP2-based RDTs two years ago. The authors should indicate what is being done in their place. This also relates to their statement on lines 453-454 that refer to the deployment of alternative malaria diagnostics.

Minor comments:

6. Figure 2 has an exceptionally small font on the Y axis that should be slightly increased. Also, I feel it would be less confusing to have the control samples listed in the same vertical order below chromosome 8 (top) and 13 (bottom) data. LC and HC should be defined in the legend.

7. The Figure 3 Y axis font is even smaller and truly impossible to read on a print out.

Reviewer #2 (Remarks to the Author):

my comments and concerns have been addressed

Author Rebuttal to Initial comments

Reviewer #1 (Remarks to the Author):

This article by Feleke et al. presents a comprehensive assessment of two rapid diagnostic tests (RDTs) employed to detect *Plasmodium falciparum* (Pf) malaria in Ethiopia. Both tests rely on

the detection of of Pf HRP2 antigen in whole blood. CareStart includes a test for *P. vivax* LDH, whereas SD Bioline includes a test for Pf LDH. The authors report the compelling finding that nearly 10% of all Pf infections sampled in Ethiopia (in their three sites) are HRP2 negative and thus would escape RDT detection. This finding was made through a powerful combination of RDT data from the two tests, PCR, molecular inversion probes and some whole-genome data. These state of the art genetic and genomic techniques, applied to a large data set of 2,714 Pf-positive samples (from a larger screen of 12,572 participants) yielded evidence that parasites with HRP2 deletions are closely related at the population level and that HRP3 deletions, which are more prevalent, occurred earlier and are more heterogeneous. The HRP2 data suggest a selective sweep arising from parasite escape of RDT screening prior to treatment. These data highlight a critical weakness in using these RDTs to detect and treat Pf malaria.

In addition to reviewing this submission, I have read the critiques from the prior submission to Nature Medicine, and feel that the authors responded very well. Overall, I find this an important study rich in data that will be of substantial interest to the malaria control and research community and other investigators studying antimicrobial diagnostics and population structures associated with intensive pathogen screening prior to treatment. I nonetheless have a few additional queries and comments:

1. Greater clarity should be provided about the way these tests work. The CareStart literature that I saw (<https://www.apacor.com/wp-content/uploads/2019/01/APA059-v8-1630-16305-CareStart-Malaria-Test-Procedure-updated-1.pdf>) indicates that one band is specific to Pf HRP2 and the other will appear positive if there is detectable LDH antigen from any of the Plasmodium species. Therefore a LDH+ band combined with a HRP2- band could indicate either a non-Pf species or Pf that lacks the hrp2 gene. The manuscript states that the CareStart is Pf HRP2 / Pv-specific LDH. The authors should clarify this apparent discrepancy. If the LDH is indeed pan specific that would change a lot of their interpretation. For the SD Bioline assay I found it surprisingly difficult to obtain information about how the assay works. Some versions on the internet show that there is only one band for Pf, which would conflate HRP2 and LDH. The manuscript indicates that the SD Bioline recognizes both Pf HRP2 and Pf-specific LDH. The authors need to clarify this. Also, they should point out that each assay has a control band. Ideally, it would help to have an additional supplementary figure that shows both types of RDTs with a description of how they work and how different banding patterns should be interpreted. The authors should also include links in their additional supplementary figure (there may be several versions of each test, leading to the apparent discrepancy above). The issue of species specific versus pan-Plasmodium is a critical one to address.

RESPONSE: Yes, we confirm that the CareStart RDT used in this survey included a *P. vivax* (not pan-species) LDH line. The provided link does not correspond to the product code used in the study (RM VM-02571). Information about this specific RDT's performance versus a standardized panel of samples can be found in the report "Malaria rapid diagnostic test performance: Summary results of WHO product testing of malaria

RDTs: round 1-8 (2008–2018)”, available at:
<https://www.who.int/publications/i/item/9789241514965>.

2. On a similar note, I think it would be very helpful to readers to have Supplementary Table 11 moved to a main Table as that shows the differences between the two assays and also documents the 332 (~9.7%) of samples that were Pf LDH+, HRP2- and Pv-LDH-. That table explicitly states that the CareStart has a Pv-specific LDH, which the authors should clarify in response to comment 1.

RESPONSE: Thank you for this suggestion. We agree that this information is important and are grateful to the reviewer for suggesting its addition during our initial revision. We regret that we are unable to include additional tables and figures in the main text due to limits specified in the Nature Microbiology publication guidelines. However, we have added an explicit reference to Supplementary Table 11 in the discussion (line 272) to highlight the availability of these results.

3. It would be helpful to know the prevalence of *P. vivax* in their cohort, even though I recognize this is not central to their study. They describe *P. vivax* data in the supplement (lines 863-878) and it would help to give a percentage (or estimate?) of the number of *P. vivax* samples as defined by their CareStart RDT. Were there any *P. vivax* mono-infections or were they always with Pf? The authors should also expand on any other evidence for *P. vivax* that they obtained using other molecular techniques (PCR? WGS?). If the Editors allow more space, it would be useful to include this section in the main manuscript.

RESPONSE: We have added a sentence to the Results section noting that 9.4% of subjects were positive for *P. vivax* by the Pf/Pv CareStart RDT (lines 135-136). This frequency is derived from Table 1: 593 (4.7%) subjects had CareStart RDT results consistent with *P. falciparum*-*P. vivax* co-infection and 590 (4.7%) had results consistent with *P. vivax* mono-infection. We did not perform *P. vivax*-specific PCR assays, and our SWGA primer sets and MIP probes were designed to target the *P. falciparum* genome.

This revision required moving sections from the main text to the supplement to comply with word limits specified in Nature Microbiology’s publication guidelines. As a result, we regret that we could not move the *P. vivax* section into the main text.

4. The authors make the worrying statement (lines 165-167) that sufficient cross-reactive HRP3 can trigger a positive HRP2 band. They then exclude hrp2- hrp3+ samples with sufficient cross-reactive HRP3. How do they determine which samples have sufficient cross-reactive HRP3? This seems like a difficult factor to correct. Any quantitative data in this regard or more explanation would be very helpful.

RESPONSE: We agree with the reviewer that it is not an easy task to determine which samples have sufficient cross-reactive HRP3. Discerning HRP2 antigenemia from HRP3 cross-reactivity is not feasible using existing RDTs or immunoassays. At present, the

general school of thought is that infection by *pfhp2-/3+* parasites with >1,000 parasites/ μ L will trigger a positive HRP2 band due to HRP3 cross-reactivity. However, this threshold is not well defined and will inevitably vary based on the antibodies used in the RDT test bands and the infecting parasite strain.

To overcome this, we focused on the prevalence of the phenotype of greatest importance to programs (false-negative HRP2-based RDT results due to infection by parasites with *pfhrp2* deletion). We did not attempt to incorporate *pfhrp3* PCR genotype into this estimate to avoid ambiguity about cross-reactivity. We include explicit discussion of our approach and its limitations in lines 170-174. Our approach provides actionable information to programs but may underestimate the true prevalence of *pfhrp2/3*-deleted parasites.

5. On page 386 the authors state that the prevalence of *hrp2/hrp3*-deleted parasites appears to have been stable in neighboring Eritrea despite removal of HRP2-based RDTs two years ago. The authors should indicate what is being done in their place. This also relates to their statement on lines 453-454 that refer to the deployment of alternative malaria diagnostics.

RESPONSE: We removed this line in the process of reducing the word count to comply with Nature Microbiology guidelines. Eritrea has transitioned to use of a Pf-LDH-based RDT. In the Discussion, we have added information about the only Pf-LDH/Pv-LDH RDT product suitable for Ethiopia that is approved for purchase using Global Fund financing (lines 273-277).

Minor comments:

6. Figure 2 has an exceptionally small font on the Y axis that should be slightly increased. Also, I feel it would be less confusing to have the control samples listed in the same vertical order below chromosome 8 (top) and 13 (bottom) data. LC and HC should be defined in the legend.

RESPONSE: Thank you for this suggestion. We have revised the font size, re-ordered the controls, and defined the LC and HC controls in the legend as suggested.

7. The Figure 3 Y axis font is even smaller and truly impossible to read on a print out.

RESPONSE: We have increased the font size to improve readability as suggested.

Reviewer #2 (Remarks to the Author):

my comments and concerns have been addressed

Decision Letter, first revision:

Dear Dr. Parr,

Thank you for submitting your revised manuscript "Plasmodium falciparum histidine-rich protein 2 deletion is under recent positive selection in Ethiopia and threatens malaria diagnostic strategies" (NMICROBIOL-21051266A). I've check the tracked version and we are now ready to proceed to finalize your manuscript for publication in Nature Microbiology, pending minor revisions to comply with our editorial and formatting guidelines.

Thank you again for your interest in Nature Microbiology. Please do not hesitate to contact me if you have any questions.

Decision Letter, final checks:

Dear Dr. Parr,

Thank you for your patience as we've prepared the guidelines for final submission of your Nature Microbiology manuscript, "Plasmodium falciparum histidine-rich protein 2 deletion is under recent positive selection in Ethiopia and threatens malaria diagnostic strategies" (NMICROBIOL-21051266A). Please carefully follow the step-by-step instructions provided in the attached file, and add a response in each row of the table to indicate the changes that you have made. Please also check and comment on any additional marked-up edits we have proposed within the text. Ensuring that each point is addressed will help to ensure that your revised manuscript can be swiftly handed over to our production team.

We would like to start working on your revised paper, with all of the requested files and forms, as soon as possible (preferably within two weeks if not before). Please get in contact with us if you anticipate delays.

In recognition of the time and expertise our reviewers provide to Nature Microbiology's editorial process, we would like to formally acknowledge their contribution to the external peer review of your manuscript entitled "Plasmodium falciparum histidine-rich protein 2 deletion is under recent positive selection in Ethiopia and threatens malaria diagnostic strategies". For those reviewers who give their assent, we will be publishing their names alongside the published article.

Nature Microbiology offers a Transparent Peer Review option for new original research manuscripts submitted after December 1st, 2019. As part of this initiative, we encourage our authors to support increased transparency into the peer review process by agreeing to have the reviewer comments, author rebuttal letters, and editorial decision letters published as a Supplementary item. When you submit your final files please clearly state in your cover letter whether or not you would like to participate in this initiative. Please note that failure to state your preference will result in delays in accepting your manuscript for publication.

Cover suggestions

As you prepare your final files we encourage you to consider whether you have any images or illustrations that may be appropriate for use on the cover of Nature Microbiology.

Nature Microbiology has now transitioned to a unified Rights Collection system which will allow our Author Services team to quickly and easily collect the rights and permissions required to publish your work. Approximately 10 days after your paper is formally accepted, you will receive an email in providing you with a link to complete the grant of rights. If your paper is eligible for Open Access, our Author Services team will also be in touch regarding any additional information that may be required to arrange payment for your article.

Please note that Nature Microbiology is a Transformative Journal (TJ). Authors may publish their research with us through the traditional subscription access route or make their paper immediately open access through payment of an article-processing charge (APC). Authors will not be required to make a final decision about access to their article until it has been accepted. Find out more

about Transformative Journals

Authors may need to take specific actions to achieve compliance with funder and institutional open access mandates. For submissions from January 2021, if your research is supported by a funder that requires immediate open access (e.g. according to Plan S principles) then you should select the gold OA route, and we will direct you to the compliant route where possible. For authors selecting the subscription publication route our standard licensing terms will need to be accepted, including our self-archiving policies. Those standard licensing terms will supersede any other terms that the author or any third party may assert apply to any version of the manuscript.

For information regarding our different publishing models please see our Transformative Journals page. If you have any questions about costs, Open Access requirements, or our legal forms, please contact ASJournals@springernature.com.

Please use the following link for uploading these materials:
{REDACTED}

{REDACTED}

Final Decision Letter:

Dear Dr Parr,

I am pleased to accept your Article "Plasmodium falciparum is evolving to escape malaria rapid diagnostic tests in Ethiopia" for publication in Nature Microbiology. Thank you for having chosen to submit your work to us and many congratulations.

Before your manuscript is typeset, we will edit the text to ensure it is intelligible to our wide readership and conforms to house style.

Acceptance of your manuscript is conditional on all authors' agreement with our publication policies

(see www.nature.com/nmicrobiolate/authors/gta/content-type/index.html). In particular your manuscript must not be published elsewhere and there must be no announcement of the work to any media outlet until the publication date (the day on which it is uploaded onto our website).

Please note that *Nature Microbiology* is a Transformative Journal (TJ). Authors may publish their research with us through the traditional subscription access route or make their paper immediately open access through payment of an article-processing charge (APC). Authors will not be required to make a final decision about access to their article until it has been accepted. [Find out more about Transformative Journals](https://www.springernature.com/gp/open-research/transformative-journals)

Authors may need to take specific actions to achieve compliance with funder and institutional open access mandates. For submissions from January 2021, if your research is supported by a funder that requires immediate open access (e.g. according to Plan S principles) then you should select the gold OA route, and we will direct you to the compliant route where possible. For authors selecting the subscription publication route our standard licensing terms will need to be accepted, including our self-archiving policies. Those standard licensing terms will supersede any other terms that the author or any third party may assert apply to any version of the manuscript.

You can now use a single sign-on for all your accounts, view the status of all your manuscript

submissions and reviews, access usage statistics for your published articles and download a record of your refereeing activity for the Nature journals.
